# MicroRNAs expression profile in CCR6$^+$ regulatory T cells

Juanjuan Zhao[1,3], Yongju Li[1,3], Yan Hu[1], Chao Chen[1], Ya Zhou[2], Yijin Tao[1], Mengmeng Guo[1], Nalin Qin[1] and Lin Xu[1]

[1] Department of Immunology, Zunyi Medical College, Guizhou, China
[2] Department of Medical Physics, Zunyi Medical College, Guizhou, China
[3] These authors contributed equally to this work.

## ABSTRACT

**Backgroud.** CCR6$^+$ CD4$^+$ regulatory T cells (CCR6$^+$ Tregs), a distinct Tregs subset, played an important role in various immune diseases. Recent evidence showed that microRNAs (miRNAs) are vital regulators in the function of immune cells. However, the potential role of miRNAs in the function of CCR6$^+$ Tregs remains largely unknown. In this study, we detected the expression profile of miRNAs in CCR6$^+$ Tregs.

**Materials and Methods.** The expression profile of miRNAs as well as genes in CCR6$^+$ Tregs or CCR6$^-$ Tregs from Balb/c mice were detected by microarray. The signaling pathways were analyzed using the Keggs pathway library.

**Results.** We found that there were 58 miRNAs significantly upregulated and 62 downregulated up to 2 fold in CCR6$^+$ Tregs compared with CCR6$^-$ Tregs. Moreover, 1,391 genes were observed with 3 fold change and 20 signaling pathways were enriched using the Keggs pathway library.

**Conclusion.** The present data showed CCR6$^+$ Tregs expressed specific miRNAs pattern, which provides insight into the role of miRNAs in the biological function of distinct Tregs subsets.

Corresponding author
Lin Xu, xulinzhouya@163.com

## INTRODUCTION

CC chemokine receptor type 6 (CCR6), a family member of chemokine receptor, was widely expressed in various immune cells (*Duhen & Campbell, 2014*; *Paradis et al., 2014*; *Wong et al., 2013*). The interaction of CCR6 and its distinct ligand CCL20 mediated the migration of immune cells into immune reaction sites (*Chen et al., 2011*; *Kallal et al., 2010*). Recent evidence showed that CCR6 was also functionally expressed on CD4$^+$CD25$^+$ regulatory T cells (Tregs) (*Rivino et al., 2010*). And CCR6$^+$ subset of Tregs, a demonstrated memory/effector phenotype, played an important role in various immune diseases (*Kitamura, Farber & Kelsall, 2010*). *Kleinewietfeld et al. (2005)* reported that CCR6$^+$ Tregs were involved in the pathogenesis of experimental allergic encephalomyelitis (EAE). In tumors, *Lamprecht et al. (2008)* reported that CCR6$^+$ Tregs might favor immune escape of Hodgkin/Reed-Sternberg (HRS) cells. Similarly, our recent work further showed that CCR6$^+$ subset of Treg cells were dominantly enriched in tumor mass and closely related

to poor prognosis of breast cancer patients (*Xu et al., 2010*). Notably, the predominant proliferation triggered by DCs was critical for their enrichment and suppressive capacity in tumor mass (*Xu et al., 2011*). However, the exact regulation mechanism involved in the biological function including proliferation and suppressive capacity of this Tregs subset remains largely unknown; it might be helpful to understanding the contribution of distinct Treg subsets to immunosuppression and ultimately aid the designing of therapy for clinical related disease.

MicroRNAs (miRNAs) are endogenous, non-coding single-stranded RNAs that are approximately 20 nucleotides in length, and have emerged as a key regulator in physiology as well as pathology attributable to its ability to downregulate gene expression through mRNA destabilization/degradation and translation repression by binding onto either $3'$ UTR of the target mRNA. Recent studies have shown that different immune cells express distinct miRNA patterns and these miRNA molecules have the ability to modify the expression of target genes and subsequently regulate the function of immune cells (*Johanson et al., 2014*; *Danger et al., 2014*; *Gigli & Maizon, 2013*). For example, miR-21 was highly expressed in CD4$^+$ T cells *(Sommers et al., 2013)*, and silencing of miR-21 could alter the proliferation and function of CD4$^+$ T cells (*Wang et al., 2014*). However, whether CCR6$^+$ Tregs also expressed specific miRNA patterns and the potential role of these miRNAs in the biological function of these cells remains to be elucidated.

To this end, in the present study, the expression pattern of miRNAs in the CCR6$^+$ Tregs was evaluated. Moreover, the mRNA expression profile which might be affected by these miRNAs also was investigated. Our data showed that CCR6$^+$ Tregs expressed distinct miRNA signatures which associated with different expression of related genes. These findings might provide novel insight in the role of miRNAs in the function of distinct subset of Tregs.

## MATERIAL AND METHODS

### Animals
Female Balb/c mice 5–6 weeks of age were purchased from the Center of Experimental Animal, Fudan University (Shanghai, China). All animals were housed in the pathogen free mouse colony at our institution and all animal experiments were performed according to the guidelines for the Care and Use of Laboratory Animals (Ministry of Health, PR China, 1998) and all experimental procedures were approved by the Zunyi Medical College Laboratory Animal Care and Use Committee (No. 20130108).

### Flow cytometry
Flow cytometry was performed on a FACSAria (BD Biosciences) with CellQuest Pro software using directly conjugated mAbs against the following human or murine markers: CD4-PerCP, CD25-allophycocyanin, and CCR6-FITC with corresponding isotype-matched controls (either BD Biosciences or eBioscience Systems). Foxp3 staining was conducted using the Murine Regulatory T cell staining kit (eBioscience) and run according to the manufacturer's protocol.

## miRNA microarray

All sample labeling and GeneChip procession were performed in Kangchen Biotech Corp (Guangzhou, China). One microgram of total RNA was labeled and then hybridized to miRCURY LNA$^{TM}$ microRNA, 7.0 arrays for 16 h at 48 °C. All washing steps were performed by a GeneChip Fluidics Station 450 and GeneChip were scanned with the GeneChip Scanner 3,000 7G. Partek was used to determine ANOVA $p$-values and fold changes for miRNAs. Data is accessible at the NCBI GEO database (accession GSE60041). Species annotations were added and used to filter only those miRNAs found in *Mus musculus*.

## Gene expression microarray

Total RNA was first converted to cDNA, followed by *in vitro* transcription to make cRNA. 5 μg of single stranded cDNA was synthesized; end labeled and hybridized, for 16 h at 45 °C, to Mouse Gene 1.0 ST arrays. All washing steps were performed by a GeneChip Fluidics Station 450 and GeneChip were scanned with the Axon GenePix 4000B microarray scanner. Partek was used to determine ANOVA $p$-values and fold changes for genes.

## Real time PCR

All reagents, primers, and probes were obtained from Applied Biosystems. A U6 endogenous control was used for normalization. Reverse transcriptase reactions and real-time PCR were performed according to the manufacturer's protocols (Applied Biosystems). RNA concentrations were determined with a NanoDrop instrument (NanoDrop Technologies). One nanogram of RNA per sample was used for the assays. All RT reactions, including no-template controls and RT minus controls, were run in triplicate in GeneAmp PCR 9700 Thermocycler (Applied Biosystems). Gene expression levels were quantified using the ABI Prism 7900HT sequence detection system (Applied Biosystems). Relative expression was calculated using the comparative threshold cycle (Ct) method. The primers used for target genes: murine miR-142a (fwd):5′-TGGCATGAGGATCAGCAGGG-3′, murine miR-142a (rev):5′–GGCAGTCCGCAGCTCTAGG-3′; murine miR-21 (fwd): 5′-GCGTGCTAATGGTGGA-3′, murine miR-21 (rev): 5′-CAGGCGTATCAGTGGG-3′.

## Statistical analyses

Statistical analyses of the data were performed with the aid of analysis programs in SPSS12.0 software. Statistical evaluation was performed using one way analysis of variance (ANOVA) or $t$ test using the program PRISM 4.0 (GraphPad Software Inc., San Diego, CA, USA). The $p$ values <0.05 were considered significant and are indicated on the figures accompanying this article as follows unless otherwise indicated: *$p < 0.05$. Unless otherwise indicated, error bars represent SD.

## RESULTS

### MicroRNA expression profiles in CCR6$^+$ Tregs

Our previous data showed that CCR6$^+$ Tregs were dominantly enriched in tumors, which was associated with their potential proliferation activity compared with their CCR6$^-$ counterpart (*Xu et al., 2010*; *Xu et al., 2011*). In order to characterize the miRNA expression profile that regulates genes involved in potential proliferation activity of CCR6$^+$ Tregs, we performed a microarray assay using Affymatrix: GeneChip miRNA 3.0 Array that contains 1,111 mouse probe sequences. Microarray assays showed that miRNA were expressed differentially in CCR6$^+$ Tregs. A total of 120 miRNA were significantly altered with the criteria of 2.0 fold change with $p < 0.05$ (Table 1). Out of the 120 altered miRNAs, 58 were upregulated in CCR6$^+$ Tregs compared with CCR6$^-$ Tregs. As shown in a pie graph of miRNA distribution based on their fold changes in expression (Fig. 1A), the majority of miRNAs altered (88 out of 120) fell into the range of 2.0–4.0 fold up or downregulation. Only eleven miRNAs (five up-regulated and another six down-regulated) displayed over 10 fold changes between two groups (Fig. 1B).

To further investigate which miRNAs was potentially involved in the proliferation activity of CCR6$^+$ Tregs, 6 miRNAs among 120 altered miRNAs, which was well documented related to the proliferation activity of T cells, was shown (Fig. 1C). In addition, we further confirmed the expression of miR-142a and miR-21 in these 6 miRNAs by quantitative PCR. Data showed that the expression of miR-142a and miR-21 were also significantly upregulated in CCR6$^+$ Tregs compared with those in CCR6$^-$ Tregs respectively (Fig. S1, $p < 0.05$), which was consistent with the data in miRNA array.

### Gene expression profile and signaling pathway in CCR6$^+$ Tregs

To investigate the possible function of these altered expression miRNA molecules in CCR6$^+$ Tregs, we detected the global gene expression changes in CCR6$^+$ Tregs. CCR6$^+$ Tregs and CCR6$^-$ Tregs were harvested and subjected to gene expression microarray assay. The altered gene expression profiles in CCR6$^+$ Tregs were shown in a heat map (Fig. 2A). Given a three-fold change and $p < 0.05$ (up and down) in differential expression as a cut-off, the number of altered genes was reduced to 1,391; 651 of them were downregulated, and 740 genes were up regulated (Tables 2 and 3).

To clarify which signaling pathways were altered in CCR6$^+$ Tregs, we applied the KEGG library and performed enrichment analysis for microarray data. Twenty signaling pathways were enriched with the criteria of 2 fold changes (Table 4), which include the inositol phosphate metabolism, T cell receptor signaling pathway, phosphatidylinositol signaling system, mTOR signaling pathway, primary immunodeficiency and some cancer signaling pathway. Some genes from those pathways were downregulated or upregulated, such as in T cell receptor signaling pathway, ICOS, ZAP70, LAT, PLC-$\gamma$1, ITK, Ras and p38 were downregulated (Fig. 3). The mTOR pathway evenly consisted of both up and downregulated genes, of which RSK, STRAD and Raptor were downregulated and PIK3c2b, TSC1 and MO25 were upregulated (Tables 2 and 3).

 

**Table 1** **120 miRNAs altered in CCR6$^+$ Tregs.**

| miRNA | Fold change | miRNA | Fold change | miRNA | Fold change |
|---|---|---|---|---|---|
| mmu-miR-30e-5p | 35.12 | mmu-miR-344d-3p | 2.35 | mmu-miR-881-3p | 0.37 |
| mmu-miR-27a-3p | 14.92 | mmu-miR-1983 | 2.34 | mmu-miR-1948-5p | 0.37 |
| mmu-miR-5117-3p | 13.35 | mmu-miR-1947-3p | 2.27 | mmu-miR-140-5p | 0.36 |
| mmu-miR-29b-3p | 11.52 | mmu-miR-3084-5p | 2.25 | mmu-miR-3080-5p | 0.35 |
| mmu-let-7a-5p | 10.21 | mmu-miR-467c-3p | 2.25 | mmu-miR-130b-3p | 0.33 |
| mmu-miR-425-5p | 8.82 | mmu-miR-3084-5p | 2.25 | mmu-miR-466e-5p | 0.32 |
| mmu-miR-29a-3p | 8.8 | mmu-miR-467c-3p | 2.25 | mmu-miR-467e-3p | 0.32 |
| mmu-miR-181a-5p | 8.43 | mmu-miR-691 | 2.24 | mmu-miR-668-5p | 0.32 |
| mmu-miR-25-3p | 5.99 | mmu-miR-691 | 2.24 | mmu-miR-24-2-5p | 0.31 |
| mmu-miR-19b-3p | 5.74 | mmu-miR-297c-5p | 2.23 | mmu-miR-467g | 0.3 |
| mmu-miR-142-3p | 5.03 | mmu-miR-1193-3p | 2.19 | mmu-let-7g-5p | 0.29 |
| mmu-miR-5105 | 4.74 | mmu-miR-767 | 2.17 | mmu-miR-669b-3p | 0.29 |
| mmu-miR-744-5p | 4.15 | mmu-miR-5625-3p | 2.14 | mmu-let-7d-3p | 0.28 |
| mmu-miR-712-5p | 3.83 | mmu-miR-673-3p | 2.13 | mmu-miR-3068-5p | 0.28 |
| mmu-let-7c-5p | 3.73 | mmu-miR-207 | 2.08 | mmu-miR-431-5p | 0.28 |
| mmu-miR-21a-3p | 3.39 | mmu-miR-670-5p | 2.07 | mmu-miR-3473b | 0.28 |
| mmu-miR-3474 | 3.37 | mmu-miR-465a-5p | 2.05 | mmu-miR-30b-5p | 0.28 |
| mmu-miR-3096b-5p | 3.27 | mmu-miR-28a-3p | 2.03 | mmu-miR-669i | 0.27 |
| mmu-miR-3470a | 3.16 | mmu-miR-1900 | 2.02 | mmu-miR-1843a-3p | 0.27 |
| mmu-miR-3097-5p | 3.07 | mmu-miR-1935 | 2.01 | mmu-miR-32-5p | 0.25 |
| mmu-miR-3097-5p | 3.07 | mmu-miR-5616-3p | 0.5 | mmu-miR-127-3p | 0.24 |
| mmu-miR-3097-5p | 3.07 | mmu-miR-881-5p | 0.5 | mmu-miR-29a-5p | 0.23 |
| mmu-miR-665-3p | 3.05 | mmu-miR-30e-3p | 0.49 | mmu-miR-669c-5p | 0.23 |
| mmu-miR-665-3p | 3.05 | mmu-miR-425-3p | 0.49 | mmu-miR-329-3p | 0.21 |
| mmu-miR-665-3p | 3.05 | mmu-miR-340-3p | 0.47 | mmu-miR-30d-5p | 0.2 |
| mmu-miR-466j | 3.03 | mmu-miR-500-3p | 0.47 | mmu-miR-3084-3p | 0.19 |
| mmu-miR-466j | 3.03 | mmu-miR-467h | 0.46 | mmu-miR-466d-5p | 0.19 |
| mmu-miR-466j | 3.03 | mmu-miR-669a-3-3p | 0.45 | mmu-miR-3962 | 0.17 |
| mmu-miR-491-3p | 3.02 | mmu-miR-669d-5p | 0.44 | mmu-miR-3069-5p | 0.17 |
| mmu-miR-466f-5p | 2.95 | mmu-miR-467f | 0.44 | mmu-miR-669p-3p | 0.16 |
| mmu-miR-5099 | 2.94 | mmu-miR-30c-5p | 0.44 | mmu-miR-3082-5p | 0.15 |
| mmu-miR-2137 | 2.94 | mmu-miR-144-3p | 0.44 | mmu-miR-423-5p | 0.14 |
| mmu-miR-26a-5p | 2.88 | mmu-miR-467e-5p | 0.44 | mmu-miR-669e-5p | 0.12 |
| mmu-miR-26b-5p | 2.84 | mmu-miR-191-5p | 0.43 | mmu-miR-374b-5p | 0.11 |
| mmu-miR-1971 | 2.74 | mmu-miR-466a/b/c/e/p-3p | 0.43 | mmu-miR-3096a-3p | 0.1 |
| mmu-miR-3473a | 2.63 | mmu-miR-665-5p | 0.42 | mmu-miR-466i-5p | 0.1 |
| mmu-miR-5129-5p | 2.61 | mmu-miR-3095-5p | 0.41 | mmu-miR-1231-3p | 0.1 |
| mmu-miR-592-3p | 2.53 | mmu-miR-466f | 0.41 | mmu-miR-467b-5p | 0.09 |
| mmu-miR-5627-5p | 2.5 | mmu-miR-511-3p | 0.38 | mmu-miR-1843b-5p | 0.06 |
| mmu-miR-33-5p | 2.44 | mmu-miR-5616-5p | 0.37 | mmu-miR-222-3p | 0.05 |

A

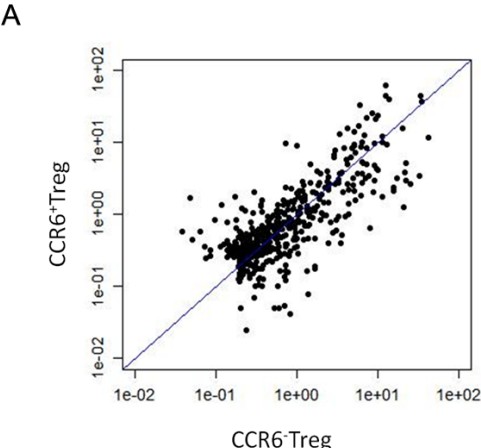

B

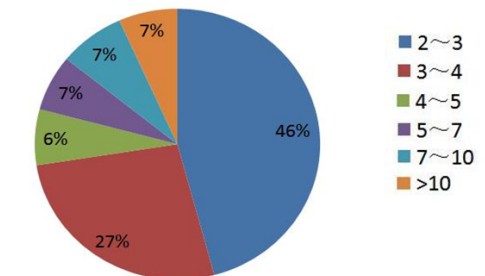

C

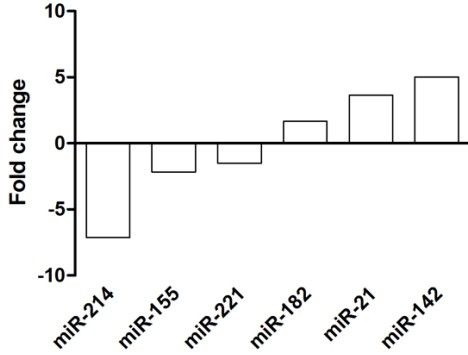

**Figure 1 miRNA expression in CCR6⁺ Tregs.** CCR6⁺ Tregs and CCR6⁻ Tregs were purified from splenocytes in Balb/c mice. The expression of miRNAs in cells was analyzed by microarray array. (A) A scatterplot of miRNA microarray. (B) A pie graph of miRNA distribution. (C) Predication of the 6 putative miRNAs associated with potential proliferation activity of CCR6⁺ Tregs based on functional similarity of target sets.

**Peer**J

A

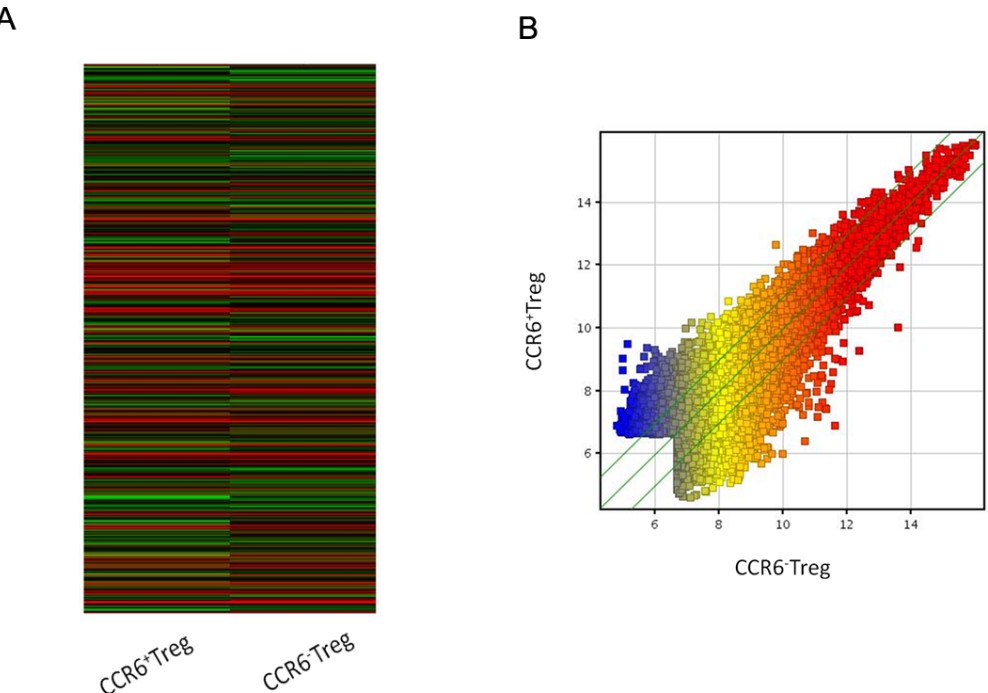

B

C

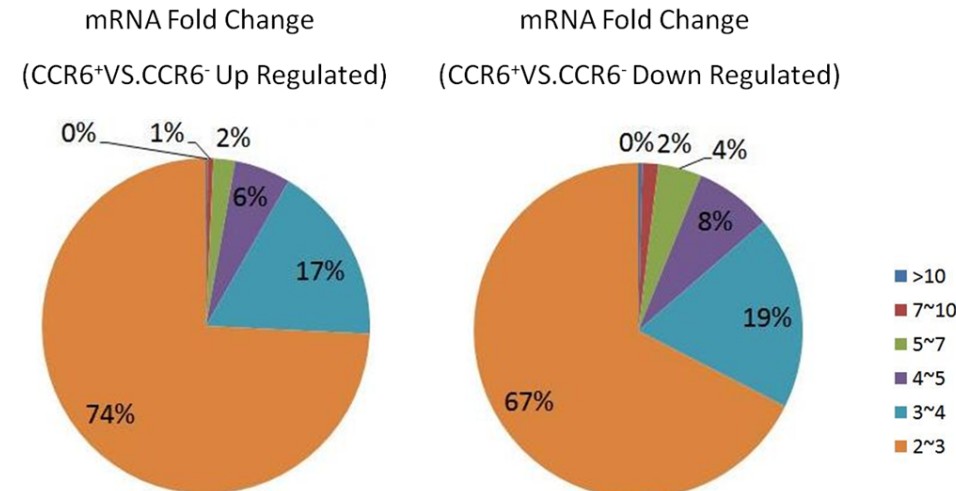

**Figure 2 Gene expression in CCR6$^+$ Tregs detected by microarray assays.** CCR6$^+$ Tregs and CCR6$^-$ Tregs were purified from splenocytes in Balb/c mice. The global expression of genes in cells was analyzed by microarray array. (A) A heat map of miRNA microarray. (B) The scatterplot for the variation between CCR6$^+$ Tregs and CCR6$^-$ Treg. (C) The fold change and frequency between CCR6$^+$ Tregs and CCR6$^-$ Tregs.

**Table 2** Over 3-fold up-regulation genes (651) in CCR6$^+$ Tregs.

| Target gene | Fold change | Target gene | Fold change | Target gene | Fold change |
|---|---|---|---|---|---|
| Kcnh7 | 21.32 | Aurkb | 3.94 | Fam195b | 3.33 |
| Olfr250 | 16.83 | Faim3 | 3.93 | Sept11 | 3.32 |
| Gm11623 | 13.12 | AU022751 | 3.93 | Chi3l3 | 3.32 |
| Treml4 | 12.86 | Igh | 3.92 | Adora3 | 3.32 |
| Dcn | 11.30 | Gda | 3.92 | Tcf7l2 | 3.32 |
| Gm13766 | 10.31 | Olfr777 | 3.92 | Sdc1 | 3.32 |
| Rpap1 | 9.22 | Gm4698 | 3.91 | Cecr2 | 3.32 |
| V1rc16 | 9.14 | Wdr66 | 3.91 | A2ld1 | 3.32 |
| Dlgap5 | 9.12 | S100a16 | 3.91 | Haao | 3.32 |
| N/A | 8.35 | Cd22 | 3.91 | AW146020 | 3.32 |
| Cts6 | 8.32 | 2610035F20Rik | 3.91 | Veph1 | 3.32 |
| Atp6v1b1 | 8.24 | Igh | 3.91 | N/A | 3.32 |
| Dnahc12 | 8.24 | Kiss1 | 3.91 | Dhdds | 3.31 |
| Adam29 | 8.17 | Brdt | 3.90 | H2-Ab1 | 3.31 |
| AW551984 | 8.11 | Pycrl | 3.89 | N/A | 3.31 |
| Hvcn1 | 8.04 | Gm2987 | 3.89 | Vmn2r38 | 3.31 |
| D630033O11Rik | 7.87 | Igh | 3.88 | 2010204K13Rik | 3.31 |
| 4933405L10Rik | 7.59 | Npepps | 3.88 | Cd72 | 3.31 |
| Igh | 7.50 | Clip2 | 3.87 | Gzmb | 3.30 |
| Fam131b | 7.23 | Gm3758 | 3.87 | Zfp385b | 3.30 |
| Ly6g5b | 7.15 | Gys1 | 3.87 | Pdgfra | 3.30 |
| Klra13 | 6.71 | N/A | 3.87 | 1700057H15Rik | 3.30 |
| Neurod6 | 6.62 | Xrcc1 | 3.87 | Txndc17 | 3.29 |
| Mef2c | 6.62 | N/A | 3.86 | Etv5 | 3.29 |
| P2ry4 | 6.43 | Zfp553 | 3.85 | Fcer1g | 3.29 |
| Neil3 | 6.39 | Nol9 | 3.85 | Gm14920 | 3.29 |
| Trappc2 | 6.39 | Tsen54 | 3.85 | Olfm4 | 3.29 |
| Tmem109 | 6.38 | Ints7 | 3.85 | N/A | 3.29 |
| Clec4n | 6.38 | Tcfeb | 3.85 | Eral1 | 3.28 |
| Vmn2r102 | 6.36 | P2ry1 | 3.84 | 2310030N02Rik | 3.28 |
| N/A | 6.36 | Hist1h2bg | 3.84 | Gm13403 | 3.28 |
| Gm10649 | 6.32 | Mxd1 | 3.84 | Idh3b | 3.28 |
| Cage1 | 6.31 | Cadps2 | 3.83 | Chd5 | 3.28 |
| Gtf2ird2 | 6.28 | N/A | 3.83 | Tssk2 | 3.28 |
| N/A | 6.27 | 2310061I04Rik | 3.83 | Cbwd1 | 3.28 |
| Eya1 | 6.22 | Fcer2a | 3.82 | Robo1 | 3.27 |
| Mpo | 6.09 | Klhl13 | 3.82 | Whsc1 | 3.27 |
| Gpr152 | 6.07 | Pah | 3.82 | Bmp1 | 3.27 |
| AI324046 | 6.04 | Zdhhc3 | 3.81 | Pygl | 3.27 |
| Ccdc82 | 5.99 | Lcn2 | 3.81 | Pvrl4 | 3.27 |
| 4933411K16Rik | 5.98 | Zbtb34 | 3.81 | Cd180 | 3.27 |
| Pigt | 5.96 | Sirpb1 | 3.81 | Tpsg1 | 3.26 |

Table 2 (*continued*)

| Target gene | Fold change | Target gene | Fold change | Target gene | Fold change |
|---|---|---|---|---|---|
| Havcr2 | 5.94 | Adam1a | 3.79 | Gprc5a | 3.26 |
| 4933402D24Rik | 5.92 | Ace2 | 3.79 | Gm13375 | 3.26 |
| Myom1 | 5.92 | C86187 | 3.79 | 1810034E14Rik | 3.26 |
| Kif2c | 5.72 | March4 | 3.79 | Il1b | 3.26 |
| Olfr514 | 5.65 | Pigq | 3.79 | C330016O10Rik | 3.26 |
| Gm7306 | 5.62 | Lingo1 | 3.78 | Ank2 | 3.25 |
| Dnajc28 | 5.59 | Nuak2 | 3.78 | Ins2 | 3.25 |
| 4930578G10Rik | 5.58 | V1rd2 | 3.77 | Hrh4 | 3.25 |
| 4930517G24Rik | 5.57 | Igh | 3.77 | Trp53rk | 3.25 |
| Gm12260 | 5.57 | Cdc20 | 3.77 | Grik1 | 3.24 |
| N/A | 5.53 | Adam9 | 3.76 | Asgr1 | 3.24 |
| Gm2847 | 5.53 | Gm13152 | 3.76 | Lrrc59 | 3.24 |
| Gp49a | 5.50 | Ccnf | 3.76 | N/A | 3.24 |
| Fcamr | 5.49 | Csgalnact2 | 3.76 | 2810408A11Rik | 3.24 |
| Klhdc7b | 5.48 | Vps53 | 3.75 | Gcet2 | 3.24 |
| Cacna1f | 5.46 | Uggt2 | 3.74 | Lrrk2 | 3.24 |
| 4930467D21Rik | 5.42 | Rbm8a | 3.73 | Pira11 | 3.24 |
| Masp1 | 5.34 | Igk | 3.73 | Tusc1 | 3.24 |
| N/A | 5.32 | Pcbp1 | 3.73 | Usp35 | 3.24 |
| Stk33 | 5.32 | Klk15 | 3.73 | Panx3 | 3.24 |
| Xirp1 | 5.31 | Smox | 3.73 | Vti1a | 3.23 |
| Prune | 5.30 | Gm5393 | 3.73 | Nudt16l1 | 3.23 |
| Brpf1 | 5.27 | Txnl4b | 3.72 | Tnk1 | 3.23 |
| Zdbf2 | 5.26 | 9130017N09Rik | 3.72 | Ighv14-2 | 3.23 |
| 4930432E11Rik | 5.24 | Rims1 | 3.72 | Hspb11 | 3.23 |
| Arhgap24 | 5.24 | Spire1 | 3.71 | Blk | 3.23 |
| N/A | 5.22 | N/A | 3.71 | Zdhhc4 | 3.23 |
| Il15 | 5.20 | Psmg4 | 3.70 | Phka1 | 3.22 |
| Plin1 | 5.18 | Mrps36 | 3.70 | Micalcl | 3.22 |
| Spink10 | 5.18 | Pstk | 3.70 | Gm13089 | 3.22 |
| Snca | 5.15 | Trmt2a | 3.70 | RP23-480B19.10 | 3.22 |
| Styxl1 | 5.14 | Nsg2 | 3.70 | Rwdd3 | 3.22 |
| Ranbp17 | 5.14 | Anxa1 | 3.70 | 1110037F02Rik | 3.22 |
| Mcam | 5.09 | Lpcat2 | 3.69 | Krtap13 | 3.21 |
| Vmn2r121 | 5.09 | Asb4 | 3.69 | Cd22 | 3.21 |
| Chi3l4 | 5.08 | Sprr2a3 | 3.69 | Hist1h2ab | 3.21 |
| Ltb4r2 | 5.02 | Rps6kb1 | 3.69 | 2700008G24Rik | 3.21 |
| Ppp1r3d | 5.02 | Zfp282 | 3.68 | N/A | 3.21 |
| Gm2705 | 5.00 | Wdfy4 | 3.68 | Chst14 | 3.21 |
| Etl4 | 4.98 | Gm2448 | 3.68 | A2bp1 | 3.20 |
| Fam108b | 4.93 | Lta4h | 3.67 | Gm2739 | 3.20 |
| Adamts8 | 4.92 | 1600020E01Rik | 3.67 | Lman1 | 3.20 |
| Akr1c13 | 4.91 | Psg29 | 3.66 | Timp1 | 3.20 |

Table 2 (*continued*)

| Target gene | Fold change | Target gene | Fold change | Target gene | Fold change |
|---|---|---|---|---|---|
| Gm11543 | 4.89 | Sik3 | 3.66 | Rad54b | 3.20 |
| Il17c | 4.89 | 4933421E11Rik | 3.65 | 1700012C08Rik | 3.20 |
| Ccdc30 | 4.89 | Ltf | 3.65 | LOC668727 | 3.20 |
| Tmed9 | 4.88 | Lpp | 3.65 | Sytl3 | 3.20 |
| Fam46a | 4.87 | H2-Aa | 3.65 | Zfp710 | 3.19 |
| N/A | 4.87 | Gm2586 | 3.64 | Pex11b | 3.19 |
| Clic5 | 4.86 | Lphn3 | 3.64 | Ncf1 | 3.19 |
| Gm5153 | 4.85 | A530023O14Rik | 3.64 | Sh3pxd2a | 3.19 |
| Fzd1 | 4.84 | Msh5 | 3.64 | Ush2a | 3.19 |
| Hemt1 | 4.82 | Gm11981 | 3.64 | Trim29 | 3.19 |
| Anxa1 | 4.79 | Crem | 3.64 | Pecam1 | 3.18 |
| Retnlg | 4.78 | Lmo2 | 3.63 | Mtus1 | 3.18 |
| Gm7219 | 4.77 | Gm4846 | 3.63 | Fam55b | 3.17 |
| Tmem63b | 4.77 | Apoo | 3.63 | Gm2461 | 3.17 |
| Clec4d | 4.75 | Btbd7 | 3.63 | Golim4 | 3.17 |
| 4933416M06Rik | 4.73 | Med8 | 3.62 | Acp1 | 3.17 |
| Zyx | 4.73 | Mgl1 | 3.62 | Gm2695 | 3.17 |
| Klk1b4 | 4.72 | Med31 | 3.62 | Kdelc2 | 3.17 |
| Defb30 | 4.71 | Abca16 | 3.61 | Myo1c | 3.17 |
| Insc | 4.65 | Hes6 | 3.61 | Gprc5b | 3.16 |
| Hs3st2 | 4.65 | Igh | 3.61 | Rcn3 | 3.16 |
| Ubap1 | 4.62 | Cdkl5 | 3.60 | Rassf4 | 3.16 |
| Gpr56 | 4.61 | Oxgr1 | 3.60 | Adrb2 | 3.16 |
| Igh-VJ558 | 4.61 | F5 | 3.60 | Cd36 | 3.16 |
| Igh | 4.61 | Psmd13 | 3.59 | Slc34a3 | 3.15 |
| Cpne2 | 4.61 | Clock | 3.59 | Acot4 | 3.15 |
| 2610028H24Rik | 4.60 | Stab1 | 3.58 | Ccdc157 | 3.15 |
| Rasl10a | 4.58 | Coasy | 3.58 | Igl-V1 | 3.15 |
| Mrpl33 | 4.58 | Fcrla | 3.57 | 4930534B04Rik | 3.15 |
| Fn3k | 4.58 | Cybb | 3.56 | Gm6127 | 3.15 |
| 9430025M13Rik | 4.57 | D2hgdh | 3.56 | 3110056O03Rik | 3.15 |
| Gm13083 | 4.55 | Igh | 3.56 | Kcnb2 | 3.15 |
| Klra6 | 4.54 | Adamtsl1 | 3.56 | Atp8b4 | 3.15 |
| 4933412E24Rik | 4.53 | BC005705 | 3.56 | Gm10883 | 3.15 |
| Zfp707 | 4.52 | Loxl4 | 3.56 | Bcr | 3.14 |
| Rapgefl1 | 4.52 | Ncapd2 | 3.55 | Mtus1 | 3.14 |
| Scyl2 | 4.50 | Hdc | 3.55 | Sgsm3 | 3.14 |
| Rab7l1 | 4.49 | Gem | 3.55 | Tdp1 | 3.14 |
| Scfd1 | 4.49 | N/A | 3.55 | Tcf15 | 3.14 |
| N/A | 4.48 | Sepx1 | 3.55 | Lmbr1 | 3.14 |
| Gm4395 | 4.48 | Tubgcp5 | 3.54 | Ermap | 3.14 |
| Odf4 | 4.46 | Cpne2 | 3.54 | 2210009G21Rik | 3.14 |
| Nfam1 | 4.46 | Rarres1 | 3.54 | N/A | 3.13 |

Table 2 (*continued*)

| Target gene | Fold change | Target gene | Fold change | Target gene | Fold change |
|---|---|---|---|---|---|
| Topbp1 | 4.46 | Ebf3 | 3.54 | Dhx35 | 3.13 |
| Grhl1 | 4.46 | Csf1r | 3.54 | Ell3 | 3.13 |
| Guf1 | 4.45 | N/A | 3.54 | 4930406D18Rik | 3.13 |
| Trpm3 | 4.44 | Igh | 3.54 | Ubd | 3.13 |
| Ciita | 4.43 | N/A | 3.54 | Gm6425 | 3.13 |
| Hist1h2ak | 4.42 | Mfsd3 | 3.53 | Hist1h3e | 3.13 |
| Igh | 4.42 | Homer2 | 3.53 | Slc22a17 | 3.13 |
| Fcgr2b | 4.42 | Zbtb16 | 3.53 | Serpinb1c | 3.12 |
| Wac | 4.42 | Ifltd1 | 3.52 | Sln | 3.12 |
| Msmb | 4.41 | Gm10693 | 3.52 | Gm10766 | 3.12 |
| Plac8 | 4.41 | Ptgs1 | 3.52 | Adipor1 | 3.12 |
| Nr5a1 | 4.38 | Sh2d3c | 3.51 | Gm684 | 3.11 |
| Gm13446 | 4.37 | V1rc29 | 3.51 | Il1f9 | 3.11 |
| Vmn2r73 | 4.37 | Lrp1 | 3.51 | Kcnj16 | 3.10 |
| Pfkfb4 | 4.37 | Nova1 | 3.51 | Car1 | 3.10 |
| Phyhipl | 4.36 | N/A | 3.51 | Psme4 | 3.10 |
| Gpatch4 | 4.36 | 4930578N18Rik | 3.49 | Siglec5 | 3.10 |
| Cenph | 4.36 | A030001D20Rik | 3.49 | N/A | 3.10 |
| Gm13154 | 4.35 | Hsf4 | 3.49 | Igk-C | 3.10 |
| Tm2d1 | 4.35 | Trem3 | 3.49 | N/A | 3.10 |
| Ptplad2 | 4.35 | Arhgap24 | 3.48 | Igh | 3.10 |
| Gm13597 | 4.34 | Lins2 | 3.48 | 2310002J15Rik | 3.09 |
| Nkd1 | 4.34 | Igh | 3.48 | G630018N14Rik | 3.09 |
| Phox2b | 4.33 | Prnd | 3.48 | Rbx1 | 3.09 |
| Cyp2j7 | 4.33 | 4930529M08Rik | 3.47 | Gm8787 | 3.08 |
| Pstpip2 | 4.31 | 3110009E18Rik | 3.47 | N/A | 3.08 |
| Fam81b | 4.29 | Hist1h2bb | 3.46 | Gm7170 | 3.08 |
| Pira3 | 4.29 | Ncapg | 3.46 | Cd19 | 3.08 |
| Gpr112 | 4.28 | E030019B13Rik | 3.46 | Wfdc1 | 3.08 |
| 5031414D18Rik | 4.27 | Gm3528 | 3.46 | Casp12 | 3.08 |
| Trpm3 | 4.27 | Gm15498 | 3.46 | 6330416G13Rik | 3.07 |
| Slco4c1 | 4.27 | Cryz | 3.46 | Il6ra | 3.07 |
| Zfp354b | 4.25 | Stard4 | 3.46 | Scd1 | 3.07 |
| Camp | 4.24 | Bfsp2 | 3.45 | H2afy | 3.07 |
| Ric3 | 4.24 | Rpap1 | 3.45 | Lmbrd1 | 3.07 |
| Tsfm | 4.23 | Vsig1 | 3.44 | Pira1 | 3.07 |
| Abcc3 | 4.22 | Olfr1431 | 3.44 | Gm5468 | 3.07 |
| BC035044 | 4.22 | Abcb4 | 3.44 | Pgap1 | 3.07 |
| C230096C10Rik | 4.22 | Vwc2 | 3.44 | Prom2 | 3.07 |
| Nkg7 | 4.20 | Rpap1 | 3.44 | Nubp1 | 3.07 |
| Gm15623 | 4.20 | 5830477G23Rik | 3.43 | C1qb | 3.07 |
| Casc1 | 4.20 | Gypa | 3.43 | Tcf7l2 | 3.06 |
| Lsm1 | 4.19 | Slc25a42 | 3.43 | Ebf1 | 3.06 |

Table 2 (*continued*)

| Target gene | Fold change | Target gene | Fold change | Target gene | Fold change |
|---|---|---|---|---|---|
| Anxa6 | 4.19 | Arhgap26 | 3.43 | Itgb6 | 3.06 |
| D130009I18Rik | 4.17 | Ccl6 | 3.42 | Terf2 | 3.06 |
| Il1b | 4.17 | Cbfa2t3 | 3.42 | Prosc | 3.06 |
| Pcdh17 | 4.16 | Snx29 | 3.42 | N/A | 3.06 |
| Clec4d | 4.16 | Ube2w | 3.42 | Il9r | 3.05 |
| Alk | 4.16 | Slc1a1 | 3.41 | Gm14206 | 3.05 |
| Cd79b | 4.15 | Olfr399 | 3.41 | Fignl1 | 3.05 |
| Zc3h7b | 4.15 | D930016D06Rik | 3.41 | Dhrs3 | 3.05 |
| Mc4r | 4.15 | Hs2st1 | 3.41 | Ikbkg | 3.05 |
| Sept8 | 4.13 | Pou3f3 | 3.41 | Map3k7ip1 | 3.05 |
| Gp49a | 4.13 | Ccdc46 | 3.41 | Lcat | 3.05 |
| N/A | 4.12 | Olfr1434 | 3.41 | Itsn1 | 3.05 |
| Smarcd1 | 4.12 | Pcdh15 | 3.40 | Creld1 | 3.05 |
| 2700050L05Rik | 4.11 | N/A | 3.40 | Gm9121 | 3.04 |
| Fmnl2 | 4.11 | Ctbp2 | 3.40 | Klrb1c | 3.04 |
| Gm11686 | 4.11 | Pla2g7 | 3.40 | Gpr116 | 3.04 |
| Ube1y1 | 4.10 | Clk2 | 3.40 | Igh-6 | 3.04 |
| 1600012P17Rik | 4.10 | Gen1 | 3.40 | Igk-C | 3.04 |
| Irf5 | 4.09 | Stoml1 | 3.39 | Cstf1 | 3.04 |
| Caskin1 | 4.08 | Prpf19 | 3.39 | Cel | 3.04 |
| Cd300lf | 4.08 | Acer2 | 3.39 | Slc30a1 | 3.04 |
| Oosp1 | 4.07 | Rhox2c | 3.39 | N/A | 3.04 |
| Xlr3a | 4.07 | Snn | 3.38 | Gm10193 | 3.03 |
| Nol4 | 4.07 | V1rb8 | 3.38 | Gm9880 | 3.03 |
| Map2k7 | 4.06 | Sema4a | 3.38 | N/A | 3.03 |
| Gm5577 | 4.05 | Tmeff1 | 3.38 | Gm2436 | 3.03 |
| Trmt12 | 4.04 | Olfr395 | 3.38 | Prr14 | 3.03 |
| Sec14l1 | 4.04 | LOC677563 | 3.38 | Spsb1 | 3.03 |
| D930015E06Rik | 4.03 | Rfc2 | 3.37 | Hbb-b2 | 3.03 |
| Slpi | 4.03 | A430075N02 | 3.37 | Acrv1 | 3.02 |
| Gga1 | 4.03 | Pvrl2 | 3.36 | Shmt1 | 3.02 |
| Tex101 | 4.03 | Snx8 | 3.36 | Bcl11a | 3.02 |
| Itsn1 | 4.02 | Adamts1 | 3.36 | N/A | 3.02 |
| Gm3323 | 4.02 | Pnmt | 3.36 | Ly6g | 3.02 |
| Gm2954 | 4.01 | Poll | 3.36 | Cd74 | 3.02 |
| Slc35e4 | 4.01 | Serpina1f | 3.35 | Fchsd2 | 3.02 |
| C1qa | 4.00 | Pla2g12a | 3.35 | Pik3cg | 3.02 |
| Retnlg | 4.00 | Kel | 3.35 | 3300005D01Rik | 3.01 |
| Cul2 | 3.99 | Cks2 | 3.35 | Prc1 | 3.01 |
| Plekhm3 | 3.99 | Axl | 3.35 | Hyou1 | 3.01 |
| Cyth2 | 3.98 | 2010110P09Rik | 3.34 | Gnb2 | 3.01 |
| Scfd2 | 3.98 | Spink12 | 3.34 | Pla2g15 | 3.01 |
| Gns | 3.98 | 4933400N17Rik | 3.33 | 2010308F09Rik | 3.01 |

Table 2 (*continued*)

| Target gene | Fold change | Target gene | Fold change | Target gene | Fold change |
|---|---|---|---|---|---|
| Yif1a | 3.96 | Cd300lf | 3.33 | Gm10270 | 3.00 |
| N/A | 3.95 | Hist1h4f | 3.33 | Pak7 | 3.00 |
| N/A | 3.95 | Zfp800 | 3.33 | C730027P07Rik | 3.00 |

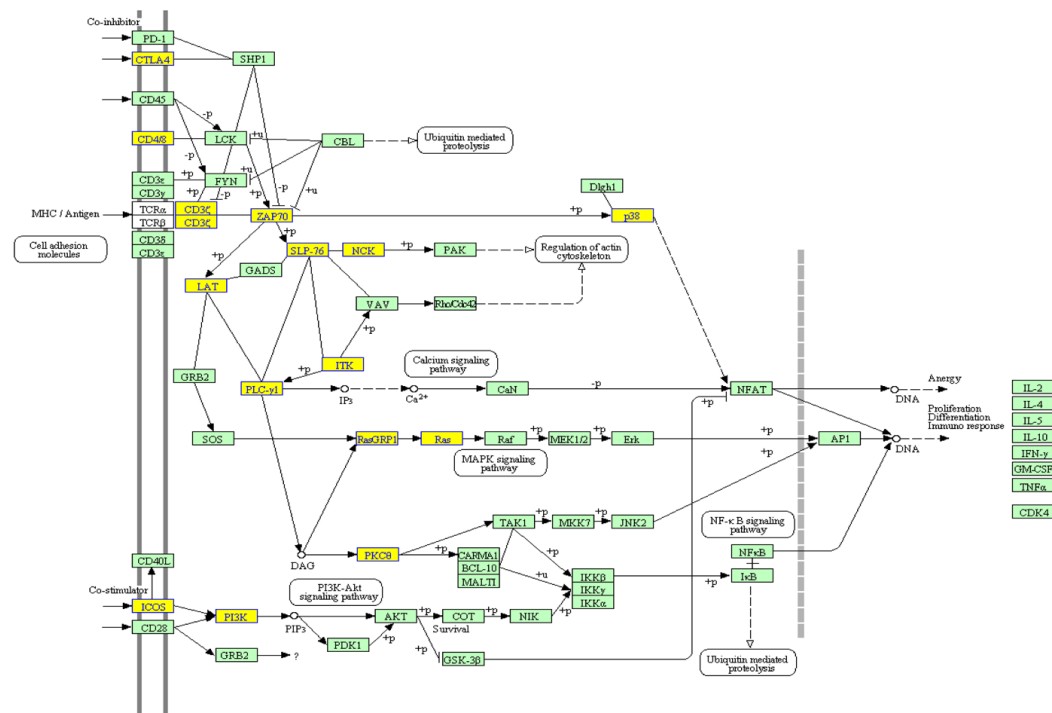

**Figure 3 Abnormal target genes of differentially expressed miRNAs were significantly enriched in the TCR signaling pathway.** The *p* value calculated by the hypergeometric distribution was set to 0.01. Downexpressed genes were shown in yellow.

## DISCUSSION

Previous studies have indicated that CD4[+]CD25[+] regulatory T cells (Treg) were a heterogeneous cell population comprising different subsets that play distinct roles in diverse animal models or human clinical disease, mediating immune suppression or immune tolerance (*Pankratz et al., 2014*; *Erfani et al., 2014*; *Lee et al., 2014*). Thus, the investigation involved in regulation of function of distinct subset of Tregs is valuable. Recent evidence showed that CCR6[+] subsets of Tregs played an important role in various immune responses such as *Villares et al.*'s (*2009*) report that CCR6[+] Tregs could inhibit the function of pathological CD4[+] Th1 cells mediated EAE pathology. We also found that CCR6[+] Tregs, but not their CCR6[−] counterpart, could dominantly enrich in tumor mass and potential inhibited the function of effector T cells *in vivo* (*Xu et al., 2010*; *Xu et al., 2011*). These findings might support the fact that CCR6[+] subset of Tregs played a critical role in tumor immunity. However, the regulation mechanism involved in the function of this subset Tregs remains largely unknown. Recent studies provided some clues to this

**Table 3** Over 3-fold down-regulation genes (740) in CCR6$^+$ Tregs.

| Target gene | Fold change | Target gene | Fold change | Target gene | Fold change |
|---|---|---|---|---|---|
| Il2ra | 25.65 | Atxn7l1 | 4.27 | Trim37 | 3.41 |
| N/A | 18.99 | Gm5282 | 4.25 | Ksr2 | 3.41 |
| Gm9119 | 15.47 | St3gal3 | 4.25 | ENSMUSG00000079376 | 3.41 |
| Il2ra | 15.05 | 4930417O13Rik | 4.24 | Ptpn5 | 3.40 |
| N/A | 14.55 | Trerf1 | 4.24 | 9230117E06Rik | 3.40 |
| Ctla4 | 14.24 | Klk6 | 4.23 | N/A | 3.40 |
| Gal3st1 | 12.37 | 2610042L04Rik | 4.22 | N/A | 3.40 |
| Gm3453 | 12.21 | Cyp4f41-ps | 4.22 | Plekha1 | 3.39 |
| Gal | 12.20 | Clcn1 | 4.21 | Trav3n-3 | 3.39 |
| ENSMUSG00000072735 | 11.93 | Abcb7 | 4.20 | Lrsam1 | 3.39 |
| Foxp3 | 11.69 | Bcs1l | 4.20 | Olfr109 | 3.39 |
| Cyb5r2 | 11.65 | Stk19 | 4.18 | Rsbn1 | 3.39 |
| Phkg1 | 10.53 | Sectm1a | 4.18 | Odf1 | 3.39 |
| Ikzf2 | 10.44 | Fmr1nb | 4.17 | Mc2r | 3.38 |
| Evc2 | 10.17 | Pnkd | 4.17 | Ifna6 | 3.38 |
| Il17rc | 10.00 | N/A | 4.17 | Gm7223 | 3.38 |
| Plekhg5 | 9.93 | Gpr110 | 4.17 | Cntn4 | 3.38 |
| ENSMUSG00000072735 | 9.66 | Inpp4b | 4.17 | N/A | 3.38 |
| Acer2 | 9.56 | Gatsl3 | 4.17 | Gm10228 | 3.38 |
| Neb | 9.55 | Dapk1 | 4.16 | Gm5169 | 3.37 |
| Gpr45 | 9.49 | Gm3455 | 4.15 | R3hcc1 | 3.37 |
| D15Wsu169e | 9.47 | Gm14717 | 4.14 | Slc38a1 | 3.37 |
| Brunol5 | 9.44 | 1700001E04Rik | 4.14 | Inpp4b | 3.37 |
| Pxdn | 9.44 | Pde4a | 4.13 | Nphp3 | 3.37 |
| Gpr83 | 9.43 | Slc35f2 | 4.13 | Csnk1g1 | 3.36 |
| ENSMUSG00000072735 | 9.43 | Adam6b | 4.13 | Jazf1 | 3.36 |
| Gm3727 | 9.36 | Penk | 4.13 | Arhgdig | 3.36 |
| Gm3727 | 9.25 | 2510048L02Rik | 4.13 | Etaa1 | 3.36 |
| N/A | 9.24 | Casp3 | 4.12 | Cul2 | 3.36 |
| Gm11744 | 9.05 | Dcaf17 | 4.12 | Gm10837 | 3.36 |
| Gm3339 | 8.66 | Gm3182 | 4.12 | Ppp2r3a | 3.36 |
| Dpy19l2 | 8.60 | 1500015O10Rik | 4.11 | Gm1574 | 3.35 |
| Caskin2 | 8.31 | Acsl4 | 4.11 | Tspan12 | 3.35 |
| Ikzf2 | 8.22 | Ddx43 | 4.10 | Magi3 | 3.35 |
| Tubgcp5 | 8.17 | AI987944 | 4.09 | 1110059M19Rik | 3.35 |
| Gm2974 | 8.16 | Plin1 | 4.09 | Cpsf4l | 3.34 |
| C230088H06Rik | 8.08 | Tox | 4.09 | Parp4 | 3.34 |
| Fbxw27 | 8.05 | Gm10338 | 4.07 | Galr3 | 3.34 |
| Gm14005 | 8.04 | Zscan12 | 4.06 | Adam33 | 3.34 |
| Gm8362 | 7.95 | Fam71e1 | 4.06 | Frs3 | 3.33 |
| Gm8297 | 7.93 | Neb | 4.06 | Ptgdr | 3.33 |
| Pla2g2d | 7.93 | 100039441 | 4.05 | BE691133 | 3.33 |

Table 3 (*continued*)

| Target gene | Fold change | Target gene | Fold change | Target gene | Fold change |
|---|---|---|---|---|---|
| Slc22a12 | 7.91 | BC106179 | 4.05 | Brp44l | 3.33 |
| N/A | 7.87 | N/A | 4.05 | Gm11468 | 3.33 |
| Cadm3 | 7.81 | Stab1 | 4.04 | Dctn4 | 3.33 |
| Cyhr1 | 7.58 | Tnfsf13b | 4.04 | E330021D16Rik | 3.33 |
| B630019K06Rik | 7.58 | Mdfi | 4.03 | Gm3764 | 3.32 |
| Inpp4b | 7.50 | A930002C04Rik | 4.03 | Cd300lg | 3.32 |
| Ctla4 | 7.49 | Slc23a3 | 4.03 | Atg2a | 3.32 |
| Cyp2u1 | 7.47 | Col6a3 | 4.02 | Ankrd9 | 3.32 |
| Gm3182 | 7.44 | Ghrh | 4.01 | Gm7225 | 3.32 |
| Tgfb2 | 7.43 | A930017M01Rik | 4.01 | Pnpla7 | 3.32 |
| Vwce | 7.41 | Itih5l | 4.01 | Cd96 | 3.31 |
| LOC100036568 | 7.32 | Aurkc | 4.00 | 4833422F24Rik | 3.31 |
| 1700029I01Rik | 7.31 | Itga6 | 4.00 | Thnsl2 | 3.31 |
| Olfr701 | 7.29 | Mfrp | 3.99 | Pdcd11 | 3.31 |
| Rfc3 | 7.29 | 1700042G15Rik | 3.99 | Robo4 | 3.31 |
| Gm10014 | 7.22 | Mageh1 | 3.98 | Aven | 3.31 |
| N/A | 7.20 | Ptpn13 | 3.98 | 1700026L06Rik | 3.31 |
| LOC100038847 | 7.16 | Olfr227 | 3.98 | Lrig2 | 3.31 |
| 544988 | 7.09 | 1700028M03Rik | 3.98 | Ehbp1 | 3.31 |
| Gm4489 | 7.07 | Gpatch4 | 3.98 | Kctd9 | 3.30 |
| LOC100038847 | 6.95 | Pxmp2 | 3.97 | Zbtb37 | 3.30 |
| Nlrx1 | 6.94 | Mllt3 | 3.97 | Lrrc34 | 3.30 |
| N/A | 6.92 | Gm10250 | 3.97 | Zfp30 | 3.30 |
| Gm3642 | 6.92 | Cux1 | 3.96 | Ano2 | 3.29 |
| Tgm1 | 6.90 | Csmd1 | 3.96 | N/A | 3.29 |
| Dmd | 6.88 | Ptger3 | 3.96 | Tmem134 | 3.29 |
| Foxp3 | 6.85 | Gm3990 | 3.95 | Sh2d6 | 3.29 |
| ENSMUSG00000072735 | 6.82 | 2010005J08Rik | 3.94 | Olfr78 | 3.29 |
| Gria1 | 6.82 | Olfr623 | 3.94 | Mapk8 | 3.29 |
| Arhgef15 | 6.81 | ENSMUSG00000072735 | 3.94 | Upp1 | 3.29 |
| Gm2888 | 6.79 | March7 | 3.94 | Gm2046 | 3.29 |
| Fdft1 | 6.73 | N/A | 3.94 | Tex21 | 3.28 |
| Gm3642 | 6.72 | Slc9a3 | 3.93 | Tnfrsf4 | 3.28 |
| Nck2 | 6.70 | Rbm9 | 3.93 | Nol11 | 3.28 |
| Adamtsl4 | 6.64 | Dtwd1 | 3.93 | 1700092C10Rik | 3.28 |
| Zfp142 | 6.60 | C77370 | 3.93 | Gm3916 | 3.28 |
| Gm3269 | 6.59 | N/A | 3.92 | Dmxl2 | 3.28 |
| Gm3411 | 6.56 | Fbxw13 | 3.92 | ENSMUSG00000072735 | 3.28 |
| 544988 | 6.53 | Amz2 | 3.92 | ENSMUSG00000079376 | 3.27 |
| 9630058J23Rik | 6.53 | Nsl1 | 3.92 | 4930587E11Rik | 3.27 |
| 2010109N18Rik | 6.51 | Plxna3 | 3.92 | Plcl1 | 3.27 |
| N/A | 6.51 | Ppme1 | 3.90 | Srgap3 | 3.27 |
| Brap | 6.51 | Gcgr | 3.90 | Prss39 | 3.27 |

| Target gene | Fold change | Target gene | Fold change | Target gene | Fold change |
|---|---|---|---|---|---|
| Tmem210 | 6.47 | Sgcd | 3.90 | Dapk3 | 3.26 |
| 4930486G11Rik | 6.46 | N/A | 3.89 | Fbxw24 | 3.26 |
| Vmn2r46 | 6.46 | ENSMUSG00000068790 | 3.89 | Gm3626 | 3.26 |
| 1110017D15Rik | 6.46 | Olfr658 | 3.88 | Mtap4 | 3.26 |
| N/A | 6.43 | Fbxo15 | 3.88 | Gm3253 | 3.25 |
| N/A | 6.43 | Mrgprb4 | 3.88 | Cypt6 | 3.25 |
| N/A | 6.38 | Ncoa7 | 3.87 | Aatf | 3.25 |
| Gm3518 | 6.36 | Grin1 | 3.87 | Il2rb | 3.25 |
| C430002E04Rik | 6.26 | 4933400A11Rik | 3.87 | Fam160a1 | 3.24 |
| Gm13620 | 6.23 | Vmn2r72 | 3.87 | Ece1 | 3.24 |
| Gm3685 | 6.23 | Pfkp | 3.87 | Nkx2-6 | 3.24 |
| Zscan10 | 6.22 | Igl | 3.86 | Pik3r2 | 3.24 |
| Gm10340 | 6.22 | 4930445K14Rik | 3.86 | Slc25a21 | 3.24 |
| Gm3159 | 6.21 | Krtap5-4 | 3.85 | Ptgfrn | 3.24 |
| 5830403L16Rik | 6.16 | Gm3424 | 3.85 | N/A | 3.24 |
| Gm3127 | 6.15 | Cd247 | 3.85 | Tbcel | 3.23 |
| B930046C15Rik | 6.13 | Samsn1 | 3.85 | Sgip1 | 3.23 |
| Syne2 | 6.10 | Uty | 3.84 | 1700023L04Rik | 3.23 |
| Gm3029 | 6.10 | Galk1 | 3.84 | Catsper3 | 3.23 |
| 1600002D24Rik | 6.06 | 1700029G01Rik | 3.84 | Dgka | 3.23 |
| Gm2224 | 6.04 | Agrn | 3.83 | 4930433N12Rik | 3.23 |
| Gm4801 | 6.00 | Lrig2 | 3.83 | Rdh16 | 3.22 |
| N/A | 5.97 | Slc25a27 | 3.83 | BC011248 | 3.22 |
| Pitpnc1 | 5.94 | Gjb4 | 3.83 | Dlgap1 | 3.22 |
| Gm3476 | 5.94 | Rgs16 | 3.83 | Olfr1283 | 3.22 |
| 6430562O15Rik | 5.92 | Cntn1 | 3.82 | Osbpl3 | 3.22 |
| Vmn2r66 | 5.92 | Fndc7 | 3.82 | Foxi2 | 3.21 |
| Gm3029 | 5.91 | Itk | 3.82 | Fam186a | 3.20 |
| Gm3115 | 5.89 | N/A | 3.82 | Gm8356 | 3.20 |
| Ndp | 5.84 | H1fx | 3.81 | Timp4 | 3.20 |
| Zfp329 | 5.83 | Pik3c2a | 3.80 | Tbc1d8 | 3.20 |
| Gpr64 | 5.82 | Ctsj | 3.80 | Srd5a1 | 3.20 |
| Nav2 | 5.81 | Emid1 | 3.80 | Olfr242 | 3.20 |
| Aven | 5.80 | Serpina1e | 3.79 | Sel1l | 3.19 |
| D030054H15Rik | 5.79 | Gm1330 | 3.79 | Mbnl2 | 3.19 |
| Grik5 | 5.76 | Tbc1d4 | 3.79 | Plac9 | 3.19 |
| Sgsm3 | 5.71 | Hs1bp3 | 3.79 | N/A | 3.19 |
| Ovol2 | 5.70 | Olfr961 | 3.79 | Slc12a1 | 3.19 |
| Mc1r | 5.65 | Pask | 3.78 | Zfp169 | 3.19 |
| Gm10371 | 5.65 | BC060267 | 3.78 | Dok7 | 3.18 |
| N/A | 5.62 | Kpna1 | 3.78 | Gm2275 | 3.18 |
| Luzp2 | 5.60 | Arg1 | 3.77 | Gm2643 | 3.18 |
| Pthlh | 5.59 | Cts8 | 3.77 | Dpep2 | 3.18 |

Table 3 (*continued*)

| Target gene | Fold change | Target gene | Fold change | Target gene | Fold change |
|---|---|---|---|---|---|
| 1700021F07Rik | 5.58 | Suclg1 | 3.77 | Pard6b | 3.18 |
| Ccbe1 | 5.56 | 1700001E04Rik | 3.77 | Cyp17a1 | 3.17 |
| Cul7 | 5.55 | Synpo2 | 3.77 | 9330111N05Rik | 3.17 |
| Cpped1 | 5.55 | 6030458C11Rik | 3.77 | Ccdc33 | 3.17 |
| Fmnl3 | 5.54 | 1190002H23Rik | 3.77 | Tub | 3.17 |
| D6Wsu163e | 5.53 | Rpusd3 | 3.76 | Rpl7l1 | 3.17 |
| Serpinb9d | 5.49 | Gm6710 | 3.76 | AW495222 | 3.17 |
| E030025P04Rik | 5.49 | Ikbkap | 3.76 | Ipcef1 | 3.17 |
| Skap1 | 5.49 | N/A | 3.76 | Tle2 | 3.17 |
| D0H4S114 | 5.47 | 4921523L03Rik | 3.75 | B3galnt2 | 3.17 |
| Piwil2 | 5.45 | Speer4f | 3.75 | Ndel1 | 3.16 |
| 4930524L23Rik | 5.45 | Gm3047 | 3.75 | Atp6v1c2 | 3.16 |
| Ykt6 | 5.43 | Synpr | 3.74 | Hnrpll | 3.16 |
| Slc24a3 | 5.40 | N/A | 3.73 | Prrg1 | 3.16 |
| N/A | 5.39 | 9030624G23Rik | 3.72 | Cyp2j13 | 3.16 |
| Gm6337 | 5.38 | Trp53inp2 | 3.72 | Espn | 3.16 |
| Gm3149 | 5.34 | 9130401M01Rik | 3.72 | Mup1 | 3.15 |
| Tnfrsf9 | 5.29 | Myst4 | 3.72 | Ptprr | 3.15 |
| Ttn | 5.28 | Gm12836 | 3.72 | Snx11 | 3.15 |
| Gpr52 | 5.27 | 2810039B14Rik | 3.71 | Chchd8 | 3.15 |
| Cntfr | 5.26 | Fastk | 3.71 | Dnm1 | 3.15 |
| ENSMUSG00000079376 | 5.25 | Inpp4b | 3.71 | Tbc1d25 | 3.15 |
| LOC100038847 | 5.22 | N/A | 3.70 | Olfr1120 | 3.14 |
| Mybpc2 | 5.22 | Prlh | 3.70 | Gm3981 | 3.14 |
| Cdon | 5.16 | Mcm8 | 3.70 | Morc2a | 3.14 |
| Slitrk6 | 5.16 | Gm15340 | 3.70 | Ttll7 | 3.14 |
| Dom3z | 5.14 | Gm4926 | 3.69 | Irf6 | 3.14 |
| Gm3149 | 5.14 | Ebpl | 3.69 | A830039H05Rik | 3.13 |
| Gm16521 | 5.14 | N/A | 3.69 | 1700024B18Rik | 3.13 |
| Smarcal1 | 5.14 | 4930417O13Rik | 3.68 | Trap1a | 3.13 |
| C230099D08Rik | 5.13 | Pcdh15 | 3.68 | Vmn2r10 | 3.13 |
| Olfr1252 | 5.11 | Ctla4 | 3.68 | Nrn1 | 3.13 |
| 4930599N23Rik | 5.11 | 4933432I09Rik | 3.68 | Mapkapk3 | 3.13 |
| Gm3642 | 5.10 | Hsd17b2 | 3.68 | 3110082J24Rik | 3.13 |
| Gm5634 | 5.09 | Fbp1 | 3.67 | Ccdc65 | 3.13 |
| Cngb1 | 5.08 | Gm5795 | 3.66 | Spag6 | 3.13 |
| Pax3 | 5.07 | Gm8159 | 3.66 | AI428936 | 3.12 |
| ENSMUSG00000068790 | 5.07 | Atf7 | 3.66 | Tiam1 | 3.12 |
| 4632404H12Rik | 5.05 | Kdm4a | 3.66 | Cenpk | 3.12 |
| Vill | 5.03 | Ocrl | 3.66 | Rapsn | 3.12 |
| Gm8050 | 5.02 | Sgol1 | 3.65 | Tm2d3 | 3.11 |
| Anks4b | 5.00 | Prox2 | 3.65 | Tiam1 | 3.11 |
| St3gal6 | 4.96 | Rnf26 | 3.65 | Tle2 | 3.11 |

Table 3 (*continued*)

| Target gene | Fold change | Target gene | Fold change | Target gene | Fold change |
|---|---|---|---|---|---|
| 1700034I23Rik | 4.96 | N/A | 3.64 | Wbp11 | 3.11 |
| Gm3172 | 4.95 | Bub1 | 3.64 | Olfr1128 | 3.11 |
| Spata18 | 4.93 | Trim63 | 3.63 | Art1 | 3.11 |
| Plcg1 | 4.92 | Slc6a9 | 3.62 | Grin3a | 3.11 |
| Has2as | 4.91 | Dst | 3.61 | 1700110K17Rik | 3.10 |
| Ntn4 | 4.90 | AI428936 | 3.61 | Bcat1 | 3.10 |
| Skap1 | 4.88 | Marveld2 | 3.60 | Iigp1 | 3.10 |
| Cyp2c50 | 4.88 | Esrrb | 3.60 | Pla2g4e | 3.10 |
| Cope | 4.87 | Gm4699 | 3.60 | Rpusd1 | 3.10 |
| N/A | 4.86 | Ttyh1 | 3.59 | Olfr638 | 3.10 |
| Gm3642 | 4.86 | Rgs16 | 3.59 | Agbl2 | 3.10 |
| Tnfrsf18 | 4.85 | 8030463A06Rik | 3.59 | 4921509O09Rik | 3.09 |
| N/A | 4.82 | 4930578E11Rik | 3.58 | Olfr389 | 3.09 |
| Snhg11 | 4.81 | Cacnb2 | 3.57 | Pcsk4 | 3.09 |
| Gm6121 | 4.81 | Setd3 | 3.57 | Pou2f1 | 3.09 |
| Ncoa7 | 4.80 | N/A | 3.57 | Brcc3 | 3.09 |
| 1700025M24Rik | 4.75 | Gm2957 | 3.56 | Gm3034 | 3.09 |
| S100a7a | 4.75 | Magea3 | 3.56 | Gm8362 | 3.09 |
| Olfr140 | 4.74 | Syngr3 | 3.56 | D030028A08Rik | 3.09 |
| Crem | 4.73 | Gm3127 | 3.55 | Fam118b | 3.08 |
| Gab3 | 4.72 | ENSMUSG00000068790 | 3.55 | Ccdc126 | 3.08 |
| Ift80 | 4.71 | Tmem176a | 3.55 | Fbxw4 | 3.08 |
| Secisbp2 | 4.69 | 1700081N11Rik | 3.55 | Cish | 3.08 |
| 1110019B22Rik | 4.67 | N/A | 3.55 | N/A | 3.08 |
| N/A | 4.66 | 9530002K18Rik | 3.54 | A630023P12Rik | 3.08 |
| Gm7750 | 4.64 | 1700008F21Rik | 3.54 | Alox12b | 3.07 |
| N/A | 4.61 | Grhl3 | 3.54 | Hsd3b4 | 3.07 |
| N/A | 4.61 | Smc2 | 3.54 | Caskin1 | 3.07 |
| Gm8026 | 4.61 | Fam46d | 3.54 | Ank3 | 3.07 |
| 4933407C03Rik | 4.61 | Mypop | 3.54 | Helz | 3.06 |
| Tmub2 | 4.59 | Spats2 | 3.53 | Taar7b | 3.06 |
| Tnfrsf25 | 4.59 | Mpa2l | 3.53 | Gm3602 | 3.06 |
| Gm3269 | 4.57 | Nosip | 3.53 | Gm10094 | 3.06 |
| Gm8297 | 4.57 | Iigp1 | 3.53 | Ptpn9 | 3.06 |
| 9130230L23Rik | 4.56 | Wdr52 | 3.51 | 1700085B03Rik | 3.06 |
| 4831440E17Rik | 4.55 | 4833442J19Rik | 3.51 | Gm7696 | 3.06 |
| N/A | 4.54 | Tiam1 | 3.51 | 2610002I17Rik | 3.06 |
| Maf | 4.54 | Snapc4 | 3.51 | Cav3 | 3.06 |
| Gm7894 | 4.54 | Dgat2 | 3.51 | Slc4a8 | 3.06 |
| 4932431H17Rik | 4.53 | Saps2 | 3.50 | Cacna2d1 | 3.06 |
| E030046B03Rik | 4.53 | Tasp1 | 3.50 | St3gal4 | 3.05 |
| Gm3264 | 4.51 | 9930013L23Rik | 3.50 | Gm5134 | 3.05 |
| Odz3 | 4.51 | Sectm1b | 3.49 | Plod2 | 3.05 |

Table 3 (*continued*)

| Target gene | Fold change | Target gene | Fold change | Target gene | Fold change |
|---|---|---|---|---|---|
| Olfr725 | 4.50 | LOC432958 | 3.49 | Gm2282 | 3.05 |
| Frmd6 | 4.49 | Grik2 | 3.49 | Rpl26 | 3.05 |
| Reck | 4.47 | B230216N24Rik | 3.49 | Ly6g6c | 3.05 |
| Cars2 | 4.47 | Pla1a | 3.49 | Gm3453 | 3.05 |
| Themis | 4.46 | Bex1 | 3.48 | Suox | 3.05 |
| Msh2 | 4.46 | N/A | 3.48 | Emilin3 | 3.05 |
| Olfr1356 | 4.45 | Slc35d1 | 3.48 | 4931422A03Rik | 3.05 |
| E030010N08Rik | 4.44 | N/A | 3.47 | Airn | 3.05 |
| Ninj2 | 4.44 | Zfp444 | 3.47 | Gm8301 | 3.04 |
| Dennd2c | 4.44 | Kcnab3 | 3.47 | Prss23 | 3.04 |
| LOC100038847 | 4.43 | Gm9893 | 3.47 | Exoc3l | 3.04 |
| Ppp2r3a | 4.42 | Afm | 3.46 | Gm3556 | 3.04 |
| Rsad1 | 4.42 | Tecpr1 | 3.46 | Car12 | 3.04 |
| Nicn1 | 4.41 | Gm7980 | 3.46 | N/A | 3.04 |
| N/A | 4.40 | V1rc26 | 3.46 | Ipcef1 | 3.03 |
| Osbpl3 | 4.38 | Pyroxd2 | 3.46 | Gm6160 | 3.03 |
| Duxbl | 4.38 | Myo1b | 3.45 | Stk30 | 3.03 |
| Olfr1019 | 4.38 | Gemin5 | 3.45 | Txk | 3.03 |
| Ripk4 | 4.37 | Dzip1 | 3.45 | Klra4 | 3.03 |
| Ermp1 | 4.37 | Pabpc3 | 3.45 | Icos | 3.03 |
| Sfmbt2 | 4.33 | Olfr781 | 3.45 | Ciapin1 | 3.02 |
| Gpt2 | 4.33 | Agrn | 3.44 | Frmd4b | 3.02 |
| Myct1 | 4.32 | Fam98c | 3.44 | Gm3278 | 3.02 |
| E330026B02Rik | 4.31 | Fam65a | 3.44 | Scrn3 | 3.02 |
| Zbtb16 | 4.31 | Plekhg1 | 3.44 | 0610031O16Rik | 3.02 |
| N/A | 4.29 | Pbld | 3.44 | Brwd2 | 3.02 |
| 2010005H15Rik | 4.29 | Epb4.1l1 | 3.43 | Numbl | 3.02 |
| Rragd | 4.28 | Zap70 | 3.43 | Raph1 | 3.02 |
| Ephb3 | 4.28 | Kcnk13 | 3.43 | N/A | 3.01 |
| Treh | 4.28 | Mrgprh | 3.43 | N/A | 3.01 |
| Krt72 | 4.28 | Gm8519 | 3.42 | Klrg1 | 3.01 |
| Snx16 | 4.28 | Cntfr | 3.42 | Srd5a1 | 3.01 |
| Tox | 4.28 | N/A | 3.41 | | |
| Il2ra | 25.65 | Atxn7l1 | 4.27 | Trim37 | 3.41 |
| N/A | 18.99 | Gm5282 | 4.25 | Ksr2 | 3.41 |
| Gm9119 | 15.47 | St3gal3 | 4.25 | ENSMUSG00000079376 | 3.41 |
| Il2ra | 15.05 | 4930417O13Rik | 4.24 | Ptpn5 | 3.40 |
| N/A | 14.55 | Trerf1 | 4.24 | 9230117E06Rik | 3.40 |
| Ctla4 | 14.24 | Klk6 | 4.23 | N/A | 3.40 |
| Gal3st1 | 12.37 | 2610042L04Rik | 4.22 | N/A | 3.40 |
| Gm3453 | 12.21 | Cyp4f41-ps | 4.22 | Plekha1 | 3.39 |
| Gal | 12.20 | Clcn1 | 4.21 | Trav3n-3 | 3.39 |
| ENSMUSG00000072735 | 11.93 | Abcb7 | 4.20 | Lrsam1 | 3.39 |

Table 3 (*continued*)

| Target gene | Fold change | Target gene | Fold change | Target gene | Fold change |
|---|---|---|---|---|---|
| Foxp3 | 11.69 | Bcs1l | 4.20 | Olfr109 | 3.39 |
| Cyb5r2 | 11.65 | Stk19 | 4.18 | Rsbn1 | 3.39 |
| Phkg1 | 10.53 | Sectm1a | 4.18 | Odf1 | 3.39 |
| Ikzf2 | 10.44 | Fmr1nb | 4.17 | Mc2r | 3.38 |
| Evc2 | 10.17 | Pnkd | 4.17 | Ifna6 | 3.38 |
| Il17rc | 10.00 | N/A | 4.17 | Gm7223 | 3.38 |
| Plekhg5 | 9.93 | Gpr110 | 4.17 | Cntn4 | 3.38 |
| ENSMUSG00000072735 | 9.66 | Inpp4b | 4.17 | N/A | 3.38 |
| Acer2 | 9.56 | Gatsl3 | 4.17 | Gm10228 | 3.38 |
| Neb | 9.55 | Dapk1 | 4.16 | Gm5169 | 3.37 |
| Gpr45 | 9.49 | Gm3455 | 4.15 | R3hcc1 | 3.37 |
| D15Wsu169e | 9.47 | Gm14717 | 4.14 | Slc38a1 | 3.37 |
| Brunol5 | 9.44 | 1700001E04Rik | 4.14 | Inpp4b | 3.37 |
| Pxdn | 9.44 | Pde4a | 4.13 | Nphp3 | 3.37 |
| Gpr83 | 9.43 | Slc35f2 | 4.13 | Csnk1g1 | 3.36 |
| ENSMUSG00000072735 | 9.43 | Adam6b | 4.13 | Jazf1 | 3.36 |
| Gm3727 | 9.36 | Penk | 4.13 | Arhgdig | 3.36 |
| Gm3727 | 9.25 | 2510048L02Rik | 4.13 | Etaa1 | 3.36 |
| N/A | 9.24 | Casp3 | 4.12 | Cul2 | 3.36 |
| Gm11744 | 9.05 | Dcaf17 | 4.12 | Gm10837 | 3.36 |
| Gm3339 | 8.66 | Gm3182 | 4.12 | Ppp2r3a | 3.36 |
| Dpy19l2 | 8.60 | 1500015O10Rik | 4.11 | Gm1574 | 3.35 |
| Caskin2 | 8.31 | Acsl4 | 4.11 | Tspan12 | 3.35 |
| Ikzf2 | 8.22 | Ddx43 | 4.10 | Magi3 | 3.35 |
| Tubgcp5 | 8.17 | AI987944 | 4.09 | 1110059M19Rik | 3.35 |
| Gm2974 | 8.16 | Plin1 | 4.09 | Cpsf4l | 3.34 |
| C230088H06Rik | 8.08 | Tox | 4.09 | Parp4 | 3.34 |
| Fbxw27 | 8.05 | Gm10338 | 4.07 | Galr3 | 3.34 |
| Gm14005 | 8.04 | Zscan12 | 4.06 | Adam33 | 3.34 |
| Gm8362 | 7.95 | Fam71e1 | 4.06 | Frs3 | 3.33 |
| Gm8297 | 7.93 | Neb | 4.06 | Ptgdr | 3.33 |
| Pla2g2d | 7.93 | 100039441 | 4.05 | BE691133 | 3.33 |
| Slc22a12 | 7.91 | BC106179 | 4.05 | Brp44l | 3.33 |
| N/A | 7.87 | N/A | 4.05 | Gm11468 | 3.33 |
| Cadm3 | 7.81 | Stab1 | 4.04 | Dctn4 | 3.33 |
| Cyhr1 | 7.58 | Tnfsf13b | 4.04 | E330021D16Rik | 3.33 |
| B630019K06Rik | 7.58 | Mdfi | 4.03 | Gm3764 | 3.32 |
| Inpp4b | 7.50 | A930002C04Rik | 4.03 | Cd300lg | 3.32 |
| Ctla4 | 7.49 | Slc23a3 | 4.03 | Atg2a | 3.32 |
| Cyp2u1 | 7.47 | Col6a3 | 4.02 | Ankrd9 | 3.32 |
| Gm3182 | 7.44 | Ghrh | 4.01 | Gm7225 | 3.32 |
| Tgfb2 | 7.43 | A930017M01Rik | 4.01 | Pnpla7 | 3.32 |
| Vwce | 7.41 | Itih5l | 4.01 | Cd96 | 3.31 |

Table 3 (*continued*)

| Target gene | Fold change | Target gene | Fold change | Target gene | Fold change |
|---|---|---|---|---|---|
| LOC100036568 | 7.32 | Aurkc | 4.00 | 4833422F24Rik | 3.31 |
| 1700029I01Rik | 7.31 | Itga6 | 4.00 | Thnsl2 | 3.31 |
| Olfr701 | 7.29 | Mfrp | 3.99 | Pdcd11 | 3.31 |
| Rfc3 | 7.29 | 1700042G15Rik | 3.99 | Robo4 | 3.31 |
| Gm10014 | 7.22 | Mageh1 | 3.98 | Aven | 3.31 |
| N/A | 7.20 | Ptpn13 | 3.98 | 1700026L06Rik | 3.31 |
| LOC100038847 | 7.16 | Olfr227 | 3.98 | Lrig2 | 3.31 |
| 544988 | 7.09 | 1700028M03Rik | 3.98 | Ehbp1 | 3.31 |
| Gm4489 | 7.07 | Gpatch4 | 3.98 | Kctd9 | 3.30 |
| LOC100038847 | 6.95 | Pxmp2 | 3.97 | Zbtb37 | 3.30 |
| Nlrx1 | 6.94 | Mllt3 | 3.97 | Lrrc34 | 3.30 |
| N/A | 6.92 | Gm10250 | 3.97 | Zfp30 | 3.30 |
| Gm3642 | 6.92 | Cux1 | 3.96 | Ano2 | 3.29 |
| Tgm1 | 6.90 | Csmd1 | 3.96 | N/A | 3.29 |
| Dmd | 6.88 | Ptger3 | 3.96 | Tmem134 | 3.29 |
| Foxp3 | 6.85 | Gm3990 | 3.95 | Sh2d6 | 3.29 |
| ENSMUSG00000072735 | 6.82 | 2010005J08Rik | 3.94 | Olfr78 | 3.29 |
| Gria1 | 6.82 | Olfr623 | 3.94 | Mapk8 | 3.29 |
| Arhgef15 | 6.81 | ENSMUSG00000072735 | 3.94 | Upp1 | 3.29 |
| Gm2888 | 6.79 | March7 | 3.94 | Gm2046 | 3.29 |
| Fdft1 | 6.73 | N/A | 3.94 | Tex21 | 3.28 |
| Gm3642 | 6.72 | Slc9a3 | 3.93 | Tnfrsf4 | 3.28 |
| Nck2 | 6.70 | Rbm9 | 3.93 | Nol11 | 3.28 |
| Adamtsl4 | 6.64 | Dtwd1 | 3.93 | 1700092C10Rik | 3.28 |
| Zfp142 | 6.60 | C77370 | 3.93 | Gm3916 | 3.28 |
| Gm3269 | 6.59 | N/A | 3.92 | Dmxl2 | 3.28 |
| Gm3411 | 6.56 | Fbxw13 | 3.92 | ENSMUSG00000072735 | 3.28 |
| 544988 | 6.53 | Amz2 | 3.92 | ENSMUSG00000079376 | 3.27 |
| 9630058J23Rik | 6.53 | Nsl1 | 3.92 | 4930587E11Rik | 3.27 |
| 2010109N18Rik | 6.51 | Plxna3 | 3.92 | Plcl1 | 3.27 |
| N/A | 6.51 | Ppme1 | 3.90 | Srgap3 | 3.27 |
| Brap | 6.51 | Gcgr | 3.90 | Prss39 | 3.27 |
| Tmem210 | 6.47 | Sgcd | 3.90 | Dapk3 | 3.26 |
| 4930486G11Rik | 6.46 | N/A | 3.89 | Fbxw24 | 3.26 |
| Vmn2r46 | 6.46 | ENSMUSG00000068790 | 3.89 | Gm3626 | 3.26 |
| 1110017D15Rik | 6.46 | Olfr658 | 3.88 | Mtap4 | 3.26 |
| N/A | 6.43 | Fbxo15 | 3.88 | Gm3253 | 3.25 |
| N/A | 6.43 | Mrgprb4 | 3.88 | Cypt6 | 3.25 |
| N/A | 6.38 | Ncoa7 | 3.87 | Aatf | 3.25 |
| Gm3518 | 6.36 | Grin1 | 3.87 | Il2rb | 3.25 |
| C430002E04Rik | 6.26 | 4933400A11Rik | 3.87 | Fam160a1 | 3.24 |
| Gm13620 | 6.23 | Vmn2r72 | 3.87 | Ece1 | 3.24 |
| Gm3685 | 6.23 | Pfkp | 3.87 | Nkx2-6 | 3.24 |

Table 3 (*continued*)

| Target gene | Fold change | Target gene | Fold change | Target gene | Fold change |
|---|---|---|---|---|---|
| Zscan10 | 6.22 | Igl | 3.86 | Pik3r2 | 3.24 |
| Gm10340 | 6.22 | 4930445K14Rik | 3.86 | Slc25a21 | 3.24 |
| Gm3159 | 6.21 | Krtap5-4 | 3.85 | Ptgfrn | 3.24 |
| 5830403L16Rik | 6.16 | Gm3424 | 3.85 | N/A | 3.24 |
| Gm3127 | 6.15 | Cd247 | 3.85 | Tbcel | 3.23 |
| B930046C15Rik | 6.13 | Samsn1 | 3.85 | Sgip1 | 3.23 |
| Syne2 | 6.10 | Uty | 3.84 | 1700023L04Rik | 3.23 |
| Gm3029 | 6.10 | Galk1 | 3.84 | Catsper3 | 3.23 |
| 1600002D24Rik | 6.06 | 1700029G01Rik | 3.84 | Dgka | 3.23 |
| Gm2224 | 6.04 | Agrn | 3.83 | 4930433N12Rik | 3.23 |
| Gm4801 | 6.00 | Lrig2 | 3.83 | Rdh16 | 3.22 |
| N/A | 5.97 | Slc25a27 | 3.83 | BC011248 | 3.22 |
| Pitpnc1 | 5.94 | Gjb4 | 3.83 | Dlgap1 | 3.22 |
| Gm3476 | 5.94 | Rgs16 | 3.83 | Olfr1283 | 3.22 |
| 6430562O15Rik | 5.92 | Cntn1 | 3.82 | Osbpl3 | 3.22 |
| Vmn2r66 | 5.92 | Fndc7 | 3.82 | Foxi2 | 3.21 |
| Gm3029 | 5.91 | Itk | 3.82 | Fam186a | 3.20 |
| Gm3115 | 5.89 | N/A | 3.82 | Gm8356 | 3.20 |
| Ndp | 5.84 | H1fx | 3.81 | Timp4 | 3.20 |
| Zfp329 | 5.83 | Pik3c2a | 3.80 | Tbc1d8 | 3.20 |
| Gpr64 | 5.82 | Ctsj | 3.80 | Srd5a1 | 3.20 |
| Nav2 | 5.81 | Emid1 | 3.80 | Olfr242 | 3.20 |
| Aven | 5.80 | Serpina1e | 3.79 | Sel1l | 3.19 |
| D030054H15Rik | 5.79 | Gm1330 | 3.79 | Mbnl2 | 3.19 |
| Grik5 | 5.76 | Tbc1d4 | 3.79 | Plac9 | 3.19 |
| Sgsm3 | 5.71 | Hs1bp3 | 3.79 | N/A | 3.19 |
| Ovol2 | 5.70 | Olfr961 | 3.79 | Slc12a1 | 3.19 |
| Mc1r | 5.65 | Pask | 3.78 | Zfp169 | 3.19 |
| Gm10371 | 5.65 | BC060267 | 3.78 | Dok7 | 3.18 |
| N/A | 5.62 | Kpna1 | 3.78 | Gm2275 | 3.18 |
| Luzp2 | 5.60 | Arg1 | 3.77 | Gm2643 | 3.18 |
| Pthlh | 5.59 | Cts8 | 3.77 | Dpep2 | 3.18 |
| 1700021F07Rik | 5.58 | Suclg1 | 3.77 | Pard6b | 3.18 |
| Ccbe1 | 5.56 | 1700001E04Rik | 3.77 | Cyp17a1 | 3.17 |
| Cul7 | 5.55 | Synpo2 | 3.77 | 9330111N05Rik | 3.17 |
| Cpped1 | 5.55 | 6030458C11Rik | 3.77 | Ccdc33 | 3.17 |
| Fmnl3 | 5.54 | 1190002H23Rik | 3.77 | Tub | 3.17 |
| D6Wsu163e | 5.53 | Rpusd3 | 3.76 | Rpl7l1 | 3.17 |
| Serpinb9d | 5.49 | Gm6710 | 3.76 | AW495222 | 3.17 |
| E030025P04Rik | 5.49 | Ikbkap | 3.76 | Ipcef1 | 3.17 |
| Skap1 | 5.49 | N/A | 3.76 | Tle2 | 3.17 |
| D0H4S114 | 5.47 | 4921523L03Rik | 3.75 | B3galnt2 | 3.17 |
| Piwil2 | 5.45 | Speer4f | 3.75 | Ndel1 | 3.16 |

Table 3 (*continued*)

| Target gene | Fold change | Target gene | Fold change | Target gene | Fold change |
|---|---|---|---|---|---|
| 4930524L23Rik | 5.45 | Gm3047 | 3.75 | Atp6v1c2 | 3.16 |
| Ykt6 | 5.43 | Synpr | 3.74 | Hnrpll | 3.16 |
| Slc24a3 | 5.40 | N/A | 3.73 | Prrg1 | 3.16 |
| N/A | 5.39 | 9030624G23Rik | 3.72 | Cyp2j13 | 3.16 |
| Gm6337 | 5.38 | Trp53inp2 | 3.72 | Espn | 3.16 |
| Gm3149 | 5.34 | 9130401M01Rik | 3.72 | Mup1 | 3.15 |
| Tnfrsf9 | 5.29 | Myst4 | 3.72 | Ptprr | 3.15 |
| Ttn | 5.28 | Gm12836 | 3.72 | Snx11 | 3.15 |
| Gpr52 | 5.27 | 2810039B14Rik | 3.71 | Chchd8 | 3.15 |
| Cntfr | 5.26 | Fastk | 3.71 | Dnm1 | 3.15 |
| ENSMUSG00000079376 | 5.25 | Inpp4b | 3.71 | Tbc1d25 | 3.15 |
| LOC100038847 | 5.22 | N/A | 3.70 | Olfr1120 | 3.14 |
| Mybpc2 | 5.22 | Prlh | 3.70 | Gm3981 | 3.14 |
| Cdon | 5.16 | Mcm8 | 3.70 | Morc2a | 3.14 |
| Slitrk6 | 5.16 | Gm15340 | 3.70 | Ttll7 | 3.14 |
| Dom3z | 5.14 | Gm4926 | 3.69 | Irf6 | 3.14 |
| Gm3149 | 5.14 | Ebpl | 3.69 | A830039H05Rik | 3.13 |
| Gm16521 | 5.14 | N/A | 3.69 | 1700024B18Rik | 3.13 |
| Smarcal1 | 5.14 | 4930417O13Rik | 3.68 | Trap1a | 3.13 |
| C230099D08Rik | 5.13 | Pcdh15 | 3.68 | Vmn2r10 | 3.13 |
| Olfr1252 | 5.11 | Ctla4 | 3.68 | Nrn1 | 3.13 |
| 4930599N23Rik | 5.11 | 4933432I09Rik | 3.68 | Mapkapk3 | 3.13 |
| Gm3642 | 5.10 | Hsd17b2 | 3.68 | 3110082J24Rik | 3.13 |
| Gm5634 | 5.09 | Fbp1 | 3.67 | Ccdc65 | 3.13 |
| Cngb1 | 5.08 | Gm5795 | 3.66 | Spag6 | 3.13 |
| Pax3 | 5.07 | Gm8159 | 3.66 | AI428936 | 3.12 |
| ENSMUSG00000068790 | 5.07 | Atf7 | 3.66 | Tiam1 | 3.12 |
| 4632404H12Rik | 5.05 | Kdm4a | 3.66 | Cenpk | 3.12 |
| Vill | 5.03 | Ocrl | 3.66 | Rapsn | 3.12 |
| Gm8050 | 5.02 | Sgol1 | 3.65 | Tm2d3 | 3.11 |
| Anks4b | 5.00 | Prox2 | 3.65 | Tiam1 | 3.11 |
| St3gal6 | 4.96 | Rnf26 | 3.65 | Tle2 | 3.11 |
| 1700034I23Rik | 4.96 | N/A | 3.64 | Wbp11 | 3.11 |
| Gm3172 | 4.95 | Bub1 | 3.64 | Olfr1128 | 3.11 |
| Spata18 | 4.93 | Trim63 | 3.63 | Art1 | 3.11 |
| Plcg1 | 4.92 | Slc6a9 | 3.62 | Grin3a | 3.11 |
| Has2as | 4.91 | Dst | 3.61 | 1700110K17Rik | 3.10 |
| Ntn4 | 4.90 | AI428936 | 3.61 | Bcat1 | 3.10 |
| Skap1 | 4.88 | Marveld2 | 3.60 | Iigp1 | 3.10 |
| Cyp2c50 | 4.88 | Esrrb | 3.60 | Pla2g4e | 3.10 |
| Cope | 4.87 | Gm4699 | 3.60 | Rpusd1 | 3.10 |
| N/A | 4.86 | Ttyh1 | 3.59 | Olfr638 | 3.10 |
| Gm3642 | 4.86 | Rgs16 | 3.59 | Agbl2 | 3.10 |

Table 3 (*continued*)

| Target gene | Fold change | Target gene | Fold change | Target gene | Fold change |
|---|---|---|---|---|---|
| Tnfrsf18 | 4.85 | 8030463A06Rik | 3.59 | 4921509O09Rik | 3.09 |
| N/A | 4.82 | 4930578E11Rik | 3.58 | Olfr389 | 3.09 |
| Snhg11 | 4.81 | Cacnb2 | 3.57 | Pcsk4 | 3.09 |
| Gm6121 | 4.81 | Setd3 | 3.57 | Pou2f1 | 3.09 |
| Ncoa7 | 4.80 | N/A | 3.57 | Brcc3 | 3.09 |
| 1700025M24Rik | 4.75 | Gm2957 | 3.56 | Gm3034 | 3.09 |
| S100a7a | 4.75 | Magea3 | 3.56 | Gm8362 | 3.09 |
| Olfr140 | 4.74 | Syngr3 | 3.56 | D030028A08Rik | 3.09 |
| Crem | 4.73 | Gm3127 | 3.55 | Fam118b | 3.08 |
| Gab3 | 4.72 | ENSMUSG00000068790 | 3.55 | Ccdc126 | 3.08 |
| Ift80 | 4.71 | Tmem176a | 3.55 | Fbxw4 | 3.08 |
| Secisbp2 | 4.69 | 1700081N11Rik | 3.55 | Cish | 3.08 |
| 1110019B22Rik | 4.67 | N/A | 3.55 | N/A | 3.08 |
| N/A | 4.66 | 9530002K18Rik | 3.54 | A630023P12Rik | 3.08 |
| Gm7750 | 4.64 | 1700008F21Rik | 3.54 | Alox12b | 3.07 |
| N/A | 4.61 | Grhl3 | 3.54 | Hsd3b4 | 3.07 |
| N/A | 4.61 | Smc2 | 3.54 | Caskin1 | 3.07 |
| Gm8026 | 4.61 | Fam46d | 3.54 | Ank3 | 3.07 |
| 4933407C03Rik | 4.61 | Mypop | 3.54 | Helz | 3.06 |
| Tmub2 | 4.59 | Spats2 | 3.53 | Taar7b | 3.06 |
| Tnfrsf25 | 4.59 | Mpa2l | 3.53 | Gm3602 | 3.06 |
| Gm3269 | 4.57 | Nosip | 3.53 | Gm10094 | 3.06 |
| Gm8297 | 4.57 | Iigp1 | 3.53 | Ptpn9 | 3.06 |
| 9130230L23Rik | 4.56 | Wdr52 | 3.51 | 1700085B03Rik | 3.06 |
| 4831440E17Rik | 4.55 | 4833442J19Rik | 3.51 | Gm7696 | 3.06 |
| N/A | 4.54 | Tiam1 | 3.51 | 2610002I17Rik | 3.06 |
| Maf | 4.54 | Snapc4 | 3.51 | Cav3 | 3.06 |
| Gm7894 | 4.54 | Dgat2 | 3.51 | Slc4a8 | 3.06 |
| 4932431H17Rik | 4.53 | Saps2 | 3.50 | Cacna2d1 | 3.06 |
| E030046B03Rik | 4.53 | Tasp1 | 3.50 | St3gal4 | 3.05 |
| Gm3264 | 4.51 | 9930013L23Rik | 3.50 | Gm5134 | 3.05 |
| Odz3 | 4.51 | Sectm1b | 3.49 | Plod2 | 3.05 |
| Olfr725 | 4.50 | LOC432958 | 3.49 | Gm2282 | 3.05 |
| Frmd6 | 4.49 | Grik2 | 3.49 | Rpl26 | 3.05 |
| Reck | 4.47 | B230216N24Rik | 3.49 | Ly6g6c | 3.05 |
| Cars2 | 4.47 | Pla1a | 3.49 | Gm3453 | 3.05 |
| Themis | 4.46 | Bex1 | 3.48 | Suox | 3.05 |
| Msh2 | 4.46 | N/A | 3.48 | Emilin3 | 3.05 |
| Olfr1356 | 4.45 | Slc35d1 | 3.48 | 4931422A03Rik | 3.05 |
| E030010N08Rik | 4.44 | N/A | 3.47 | Airn | 3.05 |
| Ninj2 | 4.44 | Zfp444 | 3.47 | Gm8301 | 3.04 |
| Dennd2c | 4.44 | Kcnab3 | 3.47 | Prss23 | 3.04 |
| LOC100038847 | 4.43 | Gm9893 | 3.47 | Exoc3l | 3.04 |

**Peer**J

Table 3 (*continued*)

| Target gene | Fold change | Target gene | Fold change | Target gene | Fold change |
|---|---|---|---|---|---|
| Ppp2r3a | 4.42 | Afm | 3.46 | Gm3556 | 3.04 |
| Rsad1 | 4.42 | Tecpr1 | 3.46 | Car12 | 3.04 |
| Nicn1 | 4.41 | Gm7980 | 3.46 | N/A | 3.04 |
| N/A | 4.40 | V1rc26 | 3.46 | Ipcef1 | 3.03 |
| Osbpl3 | 4.38 | Pyroxd2 | 3.46 | Gm6160 | 3.03 |
| Duxbl | 4.38 | Myo1b | 3.45 | Stk30 | 3.03 |
| Olfr1019 | 4.38 | Gemin5 | 3.45 | Txk | 3.03 |
| Ripk4 | 4.37 | Dzip1 | 3.45 | Klra4 | 3.03 |
| Ermp1 | 4.37 | Pabpc3 | 3.45 | Icos | 3.03 |
| Sfmbt2 | 4.33 | Olfr781 | 3.45 | Ciapin1 | 3.02 |
| Gpt2 | 4.33 | Agrn | 3.44 | Frmd4b | 3.02 |
| Myct1 | 4.32 | Fam98c | 3.44 | Gm3278 | 3.02 |
| E330026B02Rik | 4.31 | Fam65a | 3.44 | Scrn3 | 3.02 |
| Zbtb16 | 4.31 | Plekhg1 | 3.44 | 0610031O16Rik | 3.02 |
| N/A | 4.29 | Pbld | 3.44 | Brwd2 | 3.02 |
| 2010005H15Rik | 4.29 | Epb4.1l1 | 3.43 | Numbl | 3.02 |
| Rragd | 4.28 | Zap70 | 3.43 | Raph1 | 3.02 |
| Ephb3 | 4.28 | Kcnk13 | 3.43 | N/A | 3.01 |
| Treh | 4.28 | Mrgprh | 3.43 | N/A | 3.01 |
| Krt72 | 4.28 | Gm8519 | 3.42 | Klrg1 | 3.01 |
| Snx16 | 4.28 | Cntfr | 3.42 | Srd5a1 | 3.01 |
| Tox | 4.28 | N/A | 3.41 | | |

problem since they showed that miRNAs may play a regulatory role in the development and function of Tregs (*Smigielska-Czepiel et al., 2014*; *Fayyad-Kazan et al., 2012*). To gain new insight into the role of miRNAs in the function of CCR6$^+$ Tregs, differentially expressed miRNAs were investigated by microarray assay. Moreover, the regulatory pathways of putative target genes were also analyzed in CCR6$^+$ Tregs. It was found that there were significantly different miRNA expression patterns in CCR6$^+$ Tregs and CCR6$^-$ Tregs. The difference could describe one handred and twenty miRNAs, including 58 up- and 62 down-regulated, which had more than 2-fold differential expression between CCR6$^+$ Tregs and CCR6$^-$ Tregs. We speculated that the above two differences might be a reason for different functions such as proliferation activity of CCR6$^+$ Tregs compared with CCR6$^-$ Tregs.

miR-142, a distinct member of the miRNA family, is highly conserved across species and is linked to chromosome 3p22.3/12q14. Recent evidence showed that miR-142 was highly expressed in Tregs and could regulate the expansion of Tregs in response to stimulation (*Zhou et al., 2013*). In this study, we observed that miR-142 was significantly upregulated in CCR6$^+$ Tregs. Predicated by TargetScan and FINDTAR3, 14 genes were putative targets of miR-142, in which 4 genes was downregulated (Fig. S2). We also noticed that Gfi1 was downregulated in CCR6$^+$ Tregs, indicating that Gfi1 might be a target of miR-142. *Shi et al. (2013a)* and *Shi et al. (2013b)* demonstrated that Gfi1 was critical for the development

**Table 4  KEGG pathways annotation of abnormal miRNA target genes.**

| Pathway | MAPP name | Enrichment Score |
| --- | --- | --- |
| mmu00562 | Inositol phosphate metabolism | 3.988221 |
| mmu04070 | Phosphatidylinositol signaling system | 3.533671 |
| mmu05410 | Hypertrophic cardiomyopathy (HCM) | 2.394271 |
| mmu04725 | Cholinergic synapse | 2.227839 |
| mmu05412 | Arrhythmogenic right ventricular cardiomyopathy (ARVC) | 2.126784 |
| mmu04724 | Glutamatergic synapse | 2.109772 |
| mmu03460 | Fanconi anemia pathway | 2.017738 |
| mmu05142 | Chagas disease (American trypanosomiasis) | 2.010757 |
| mmu04150 | mTOR signaling pathway | 1.906663 |
| mmu04660 | T cell receptor signaling pathway | 1.713143 |
| mmu05322 | Systemic lupus erythematosus | 12.6937 |
| mmu04640 | Hematopoietic cell lineage | 6.723747 |
| mmu05034 | Alcoholism | 6.20107 |
| mmu05152 | Tuberculosis | 5.152889 |
| mmu04662 | B cell receptor signaling pathway | 4.675411 |
| mmu05202 | Transcriptional misregulation in cancer | 4.643977 |
| mmu04672 | Intestinal immune network for IgA production | 4.281526 |
| mmu04380 | Osteoclast differentiation | 4.255375 |
| mmu05150 | Staphylococcus aureus infection | 3.867061 |
| mmu05340 | Primary immunodeficiency | 3.857659 |

**Notes.**

[a] Gray indicates downregulated target genes in the KEGG pathway.

[b] In differentially expressed genes, 15 miRNA target genes were enriched into T cell receptor (TCR) signaling pathway (Fig. 3).

of Tregs. Moreover, loss of Gfi-1 could endow the aberrant expansion of Tregs through IL-2 signaling pathway. Thus, further study on miR-142 function will help us understand the regulatory role of miR-142 in the function and proliferation of CCR6$^+$ Tregs.

MiR-21 is one of the first discovered miRNAs that is known to be widespread in human tissues such as heart, lung, brain and liver. Further data has highlighted that miR-21 can regulate the biological character of various cells including survival, invasion and apoptosis (*Shi et al., 2013a*; *Shi et al., 2013b*; *Bullock et al., 2013*; *Niu et al., 2012*). In particular miR-21 was documented as an important regulator actor in the proliferation of cells. For example, *Liu et al. (2014)* reported that miR-21 could enhance the proliferation of cancer cells through the Akt pathway. Interestingly, recent evidence further showed that miR-21 was also functionally expressed in T cells (*Sommers et al., 2013*), miR-21 could support survival of CD4$^+$ T cells, and was an important signature in CD4$^+$ T cells proliferation. Also, silencing of miR-21 could alter the proportion of CD4$^+$ T cells in lupus mice (*Wang et al., 2014*). Consistently, we observed an increase in the expression of miR-21 in CCR6$^+$ Tregs. Therefore, further study on the possible role of miR-21 was also valuable for the understanding of proliferation of CCR6$^+$ Tregs.

The data from gene expression microarray showed that 1,391 genes (651 downregulated and 740 up-regulated) significantly changed more than three fold in CCR6$^+$ Tregs. Among them, some genes have been demonstrated to be involved in the proliferation and function of Tregs. For example, TCR signaling pathway was critical for the proliferation and function of CCR6$^+$ Tregs. We noticed that some genes, including ZAP70, LAT and PLC-1 were downregulated, indicating weak transduction of TCR signaling pathway in CCR6$^+$ Tregs. Consistently, previous literature showed that CCR6$^+$ Tregs demonstrated a memory/effector phenotype and more sensitivity for the stimulation of TCR (*Kleinewiet-feld et al., 2005*). In addition, *Hanschen et al. (2012)* reported that TCR stimulation could induce rapid and higher activation of ZAP70 in Tregs, indicating that phosphorylation of ZAP70 also might be important for the proliferation of CCR6$^+$ Tregs. Therefore, these altered genes might be good targets for the proliferation and function of CCR6$^+$ Tregs in successive research work. In addition, we would point out that we did not find any prominently annotated biological category using miRNA-mRNA anti-correlations in the present study. We propose this reflects the fact that the interaction of miRNA and mRNA in the biology of CCR6$^+$ Tregs is complex which remains to be elucidated in future work.

In summary, to our knowledge, this is the first time that CCR6$^+$ Tregs, a distinct subset of Tregs, has been shown to express a distinct miRNA profile; this will help us to understand the potential role of miRNAs in the biological function of CCR6$^+$ Tregs.

### Funding

This work was supported by the National Natural Science Foundation of China (81260398, 31370918), the Program for New Century Excellent Talents in University, Ministry of Education of China (NCET-12-0661), the International Cooperation Foundation of Guizhou Province (2010-7031), the Specific Foundation for the Scientific Educational Talent of President of Guizhou Province (09C457), the Project of Guizhou Provincial Department of Science and Technology (2009C491) and the Zunyi Medical College Start-up Fund (2008F-329). The funders had no role in study design, data collection and analysis, decision to publish, or preparation of the manuscript.

### Grant Disclosures

The following grant information was disclosed by the authors:
National Natural Science Foundation of China: 81260398, 31370918.
New Century Excellent Talents in University, Ministry of Education of China: NCET-12-0661.
International Cooperation Foundation of Guizhou Province: 2010-7031.
Specific Foundation for the Scientific Educational Talent of President of Guizhou Province: 09C457.
Guizhou Provincial Department of Science and Technology: 2009C491.
Zunyi Medical College Start-up Fund: 2008F-329.

## Competing Interests

The authors declare there are no competing interests.

## Author Contributions

- Juanjuan Zhao analyzed the data, wrote the paper, reviewed drafts of the paper.
- Yongju Li performed the experiments, analyzed the data, prepared figures and/or tables, reviewed drafts of the paper.
- Yan Hu performed the experiments, analyzed the data, contributed reagents/materials/analysis tools, prepared figures and/or tables, reviewed drafts of the paper.
- Chao Chen analyzed the data, reviewed drafts of the paper.
- Ya Zhou performed the experiments, contributed reagents/materials/analysis tools, prepared figures and/or tables, reviewed drafts of the paper.
- Yijin Tao performed the experiments, analyzed the data, wrote the paper, prepared figures and/or tables, reviewed drafts of the paper.
- Mengmeng Guo performed the experiments, analyzed the data, contributed reagents/materials/analysis tools, wrote the paper, prepared figures and/or tables, reviewed drafts of the paper.
- Nalin Qin performed the experiments, wrote the paper, prepared figures and/or tables, reviewed drafts of the paper.
- Lin Xu conceived and designed the experiments, performed the experiments, wrote the paper, prepared figures and/or tables, reviewed drafts of the paper.

## Animal Ethics

The following information was supplied relating to ethical approvals (i.e., approving body and any reference numbers):

Zunyi Medical College Laboratory Animal Care and Use Committee (No.20130108).

## Microarray Data Deposition

The following information was supplied regarding the deposition of microarray data:

NCBI GEO database accession: GSE60041.

## Supplemental Information

Supplemental information for this article can be found online at http://dx.doi.org/10.7717/peerj.575#supplemental-information.

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
