# Peer review of "MicroRNAs expression profile in CCR6+ regulatory T cells"

_PeerJ, doi:10.7717/peerj.575_

## Round 0.1 · original submission · Major Revisions

It is very important that you justify very clearly the importance and significance of the miRNAs that you have selected and how/why you focused on the specific targets that you have chosen.

Reviewer 1 ·

Basic reporting

This research is an extension of the findings published by the same group. The design of experiment is good. And, the results were clear and analyzed well, which might be interesting for researchers in related field.

Experimental design

The design of experiment is good

Validity of the findings

The results were clear and analyzed well,

Additional comments

Comments to the Author
The research paper, “MicroRNAs expression profile in CCR6+ regulatory T cells” by Zhao et al. offers an insight of miRNA role in distinct regulatory T cell subset. Data showed that CCR6+ regulatory T cells expressed different miRNAs profile and gene profile compared with their CCR6- counterpart. This research might be interesting for researchers in related field. There are some minor points that need to be solved by the authors:

Minor comments:
1. The GEO number of miRNA expression assay should be supplied.
2. The statistical methods are not very clear and should be verified in Supplementary data fig 1. And the exact p-values also should be listed.

Reviewer 2 ·

Basic reporting

The study is too descriptive.

Experimental design

In general, poor quality of experimental design.

Validity of the findings

Poor

Additional comments

In this study, Zhao et al we examined the expression profile of miRNAs in CCR6+ Tregs. They found that 58 upregulated and 62 downregulated miRNAs at least 2-fold in CCR6+Tregs as compared to CCR6-Tregs. This paper is too descriptive and there appears to be no logic in selecting the miRNAs for validation. Moreover, the data in the paper is of poor quality and without proper validation and some functional significance. Specific comments are listed below.

1. miR-142 and miR-21 upregulation in CCR+Tregs was validated by qRT-PCR. Error bars and p-value are missing. Also, in the text the authors mention miR-142a whereas in Figure S1, they validate miR-142. Did they validate miR-142a or miR-142b?

2. What was the rationale for selecting miR-21 and miR-142 for further validation? In other words, why did the authors select these 2 miRNAs out of the 58 miRNAs?

3. Y-axis label in Figure S2 is missing. Error bars and p-value is also missing for this graph. This is important because some of the differences appear very marginal.

4. In Figure S2, the mRNA levels of putative targets of miR-142 were measured. Some targets showed inverse correlation.

5. In the text supplemental Figures are cited as FigS1 whereas in the Figure they are shown as SFigure1. This should be corrected.

6. Figure S2 is missing from the results.

7. How did the authors select the targets of miR-142a for validation in Figure S2? Which bioinformatic tool was used? Putative binding sites should have been shown.

---

## Round 0.2 · accepted · Accept

The reviewers and I have found that your revisions meet their concerns and I am happy to inform you that your manuscript is now acceptable for publication.

Reviewer 1 ·

Basic reporting

The resubmitted research paper, “MicroRNAs expression profile in CCR6+ regulatory T cells” by Zhao et al. fully addressed my question.

Experimental design

The designe of experiment is well. And the methods was described well.

Validity of the findings

The data is controlled and available in acceptable discipline-specific repository. The conclusioni is well stated.

Additional comments

No.

Reviewer 2 ·

Basic reporting

Good

Experimental design

Good

Validity of the findings

Good

Additional comments

The authors have responded to my critiques.